# Light-Triggered Polymersome-Based Anticancer Therapeutics Delivery

**DOI:** 10.3390/nano12050836

**Published:** 2022-03-02

**Authors:** Elisa Hernández Becerra, Jennifer Quinchia, Cristina Castro, Jahir Orozco

**Affiliations:** 1Max Planck Tandem Group in Nanobioengineering, Institute of Chemistry, Faculty of Natural and Exact Sciences, University of Antioquia, Complejo Ruta N, Calle 67 No. 52-20, Medellín 050010, Colombia; elisa.hernandez@udea.edu.co (E.H.B.); jennifer.quinchia@udea.edu.co (J.Q.); 2Engineering School, Pontificia Bolivariana University, Bloque 11, Cq. 1 No. 70-01, Medellín 050004, Colombia; cristina.castro@upb.edu.co

**Keywords:** polymersomes, light, anticancer drugs, delivery, nanocarriers

## Abstract

Polymersomes are biomimetic cell membrane-like model structures that are self-assembled stepwise from amphiphilic copolymers. These polymeric (nano)carriers have gained the scientific community’s attention due to their biocompatibility, versatility, and higher stability than liposomes. Their tunable properties, such as composition, size, shape, and surface functional groups, extend encapsulation possibilities to either hydrophilic or hydrophobic cargoes (or both) and their site-specific delivery. Besides, polymersomes can disassemble in response to different stimuli, including light, for controlling the “on-demand” release of cargo that may also respond to light as photosensitizers and plasmonic nanostructures. Thus, polymersomes can be spatiotemporally stimulated by light of a wide wavelength range, whose exogenous response may activate light-stimulable moieties, enhance the drug efficacy, decrease side effects, and, thus, be broadly employed in photoinduced therapy. This review describes current light-responsive polymersomes evaluated for anticancer therapy. It includes light-activable moieties’ features and polymersomes’ composition and release behavior, focusing on recent advances and applications in cancer therapy, current trends, and photosensitive polymersomes’ perspectives.

## 1. Introduction

Biomimetic nanostructures, such as polymersomes, also known as polymeric vesicles, are self-assembled supramolecular organizations with emulating functionalities of biological processes of permeable cell membranes [1,2]. The supramolecular architectures result from intermolecular interactions between molecules or building blocks conforming to intermolecular bonds. The non-covalent interactions among molecules propitiate molecular recognition and self-assembly processes [3,4]. There are plenty of types of supramolecular assemblies for therapeutic applications, such as host–guest structures [5,6,7], organic–inorganic hybrid materials [8,9,10], or metal-coordination complexes [11,12]. Even biological mechanisms and functions have inspired supramolecular chemistry and its complex assemblies. Biological components, such as lipids, are organized through self-assembly, resulting in the cellular bilayer [3].

Hence, imitating natural functions, such as cell compartmentalization, represents a valuable strategy to face a variety of engineering and materials science challenges [13]. Among the supramolecular architectures, the features of polymersomes make them amenable for simultaneously encapsulating both hydrophilic and hydrophobic cargoes in their core and membrane, respectively, and functionalizing their surface for cell recognition and targeted transport of therapeutic agents [14,15,16]. It is achievable, due to the overexpression of specific biomarkers on the cell membrane that interact with functionalized polymeric vesicles, for cargo uptake and release, cell imaging, diagnosis, and theragnostic duties [16,17,18,19,20]. Additionally, the great variety of polymers in type, composition and molecular weight, and formation methodologies contribute to tuning the physicochemical properties of the resulting polymersomes [18,21].

Polymersomes are, therefore, suitable platforms for the encapsulation and targeted delivery of therapeutic agents to fight pathologies such as cancer, which was the second leading cause of death globally, with 10 million deaths in 2020 [22]. New alternatives to traditional therapies are necessary to overcome drawbacks such as the low specificity and selectivity, chemoresistance, and drug short half-life [23,24,25]. Aiming to solve these issues, (nano)carriers based on supramolecular structures for cancer therapy [4,26], e.g., the polymersomes, emerge as a promising approach considering directed drug cargo release as an enhanced therapeutic [21]. Therefore, mimicking natural vesicles’ advantageous characteristics, cost-affordable bottom-up polymersomes have been nanoengineered as versatile compartmentalized systems for drug delivery [2,15,27] compared to standard top-down methodologies.

Beyond the protection and targeted delivery of therapeutic payloads by the compartmentalized structures, their function can be activated and boosted by external or internal stimuli that are conducive to an “on-demand” release of the active components, becoming accreditor of the “smart” adjective in the last decade [28]. Among external stimuli, such as ultrasound, temperature, or electric field, light constitutes a spatial-temporal, innocuous, controllable stimulus in broad wavelengths ranging from ultraviolet (UV) to near-infrared (NIR) light. Hence, electromagnetic radiation can trigger photochemical reactions, leading to the destabilization of the structures, followed by the release of the payloads or even activating light-sensitive moieties for therapeutic purposes [29].

This review aims to describe current light-triggered polymersomes evaluated for cancer therapeutic agent delivery and anticancer therapy. It first discusses the design and synthesis of polymersomes to focus, then, on polymersomes as a versatile platform for stimuli-responsive anticancer therapeutic delivery triggered explicitly by light. Hereafter, the primary mechanisms for the photoactivation of photosensitive systems are described through a systematic revision and the last years’ polymersome systems activated by light. Finally, recent findings and current trends on photosensitive (nano)carrier and (nano)reactor perspectives are pointed out, emphasizing their impact on future cancer therapeutics.

## 2. Influence of Biological Barriers on Drug Delivery Systems

Drug Delivery Systems (DDSs) are engineered platforms used as carriers of therapeutic drugs/agents into the patient’s body [30] to overcome the side effects and/or limitations associated with the administration of free therapeutic agents. Chemotherapeutic agents, for example, often have low selectivity, i.e., they kill fast-growing cells, including both tumor and healthy cells (hair, intestinal epithelial cells, or bone marrow) [31], and present limited efficiency due to the development of multidrug resistance (MDR) [31,32] or premature degradation [32]. Those based on hydrophilic biomacromolecules (e.g., DNA, siRNA, micro-RNA, proteins, and peptides) have a short half-life due to proteolytic and/or hydrolytic degradation, rapid clearance by the mononuclear phagocyte system (MPS), and filtration by the kidneys [32]. Besides, these biomacromolecules are not efficiently taken up by cancer cells because of their inability to cross the bilayer of the lipid-rich cell membrane [32].

Nanoparticle-based DDSs (NP-DDSs) are widely investigated as nanocarriers for cancer treatment due to their advantages over other DDSs (e.g., microparticle-based DDSs) and free anticancer therapeutic agents. Nanocarriers have at least one size dimension in the order of 1–100 nm, a scale in which molecules of life, such as nucleic acids, proteins, and other chemicals, fall right into [31,33,34]. However, nanocarriers with dimensions smaller than 500 nm are occasionally accepted because their biochemical and physiochemical properties are easily modifiable and compatible with DDS applications [35,36]. NPs may store therapeutic agents inside by entrapment, surface covalent conjugation, surface adsorption, or encapsulation [31,37], and the amount of therapeutic agent depends on the type, chemical composition, and the architecture of the NPs, NP–therapeutic agent interactions, among other parameters.

The potential of NPs as DDSs, including lipidic, polymeric, or inorganic NPs, has been wildly reviewed [36,37,38,39,40,41,42,43,44,45,46]. Lipid-based NPs include a subset of spherical structures (liposomes and lipidic NPs), where one lipid bilayer surrounds at least one internal aqueous compartment. The advantages of lipid-based NPs include simple formulation, self-assembly, biocompatibility, high bioavailability, and payload flexibility for hydrophilic and hydrophobic cargo encapsulation. However, lipid-based NPs have low cargo-retention efficiency and are quickly degraded and cleaned in the liver and spleen [45]. Polymer-based NPs are assembled from natural or synthetic polymers through emulsification, nanoprecipitation, ionic gelation, and microfluidics, among other methods. According to the type of polymer and the assembly methodology, a wide variety of spherical structures with high chemical versatility can be achieved, including polymersomes, polymeric micelles, and dendrimers. The chemical versatility enables hydrophilic and hydrophobic cargo delivery and easy superficial modification. Polymer-based NPs are ideal candidates for DDSs because they are biodegradable, biocompatible, water-soluble, stable, and have high cargo-retention efficiency [47,48,49]. The main disadvantage of polymer-based NPs is the risk of particle aggregation and toxicity [45]. Inorganic NPs include gold NPs (AuNPs), magnetic iron oxide NPs (IONPs), quantum dots (QDs), and mesoporous silica NPs (MSNPs) of the widest variety of sizes and shapes (e.g., nanospheres, nanorods, nanostars, nanocages, and nanoshells). While only MSNPs are used as DDSs, the others are involved in diagnostic and imaging applications due to their magnetic, radioactive, or plasmonic properties [45]. A particular class of NP-DDSs involves inorganic–organic hybrids. Metal-organic frameworks (MOFs) comprise organic ligands (e.g., carboxylates, phosphonates, imidazolates, and phenolates) and metal ions/metal clusters (e.g., transition metal and lanthanide metal) via coordinative bonds into one-, two-, or three-dimensional networks [50,51]. MOFs are highly porous and crystalline materials with remarkable cargo loading capacity, including small drug molecules, peptides, and even biomacromolecules, with efficiencies sometimes close to 100% [51].

Overall, NPs offer many attractive advantages, including (i) solubilization of hydrophobic therapeutic agents; (ii) prevention of the premature interaction of therapeutic agents with the biological environment (e.g., highly acidic environment in the stomach or the lysosomes into cells and high levels of proteases or other enzymes in the bloodstream) and therefore their degradation; (iii) control of the pharmacokinetic profile; (iv) increasing therapeutic agent concentration in the tumor site, enhancing absorption of the drug into the tumor, or improving intracellular uptake; (v) co-delivery of multiple therapeutic and/or diagnostic agents for theragnostic applications; and (vi) controlled release [32,33,46,52]. Although NPs have significant advantages by themselves as DDSs, it is essential to understand that factors such as chemical composition, size, shape, and surface properties (charges, functional groups, and ligands) determine the journey of nanocarriers in the circulation (including clearance), toxicity at the organ and system levels, nanocarrier transport, half-life, and targeting efficiency.

A summary of the biochemical and physicochemical properties of NPs for enhanced DDSs is illustrated in Figure 1.

### 2.1. Nanocarrier Stability and Clearance inside Biological Environments

Once nanocarriers are in the circulatory system, they are immediately subjected to clearance mainly, by the MPS (liver, spleen, and bone marrow), the renal system, and the immune and complement systems, depending on the NP size and surface properties [32,45]. For example, nanocarriers smaller than 8 nm are filtered out and cleared by the kidneys. The liver rapidly clears nanocarriers with sizes between 10 and 20 nm and even between 10 and 150 nm. The spleen preferentially clears nanocarriers larger than 200 nm [32]. Depending on the charge, opsonization (i.e., the adsorption of serum proteins) occurs on the surface of nanocarriers, making them more visible to the phagocytic cells in the MPS organs [32,45]. The opsonization is quite frequent in positively charged nanocarriers and they, therefore, are rapidly cleared by the MPS, followed by negatively charged nanocarriers with a longer circulation half-life than their positive counterparts. Neutral nanocarriers have a “stealth” capability, i.e., have the least protein adsorption (a significant reduction in clearance by the MPS) and the most prolonged circulation half-life. By choosing an appropriate size and surface chemistry, nanocarriers can evade clearance and have a longer circulation half-life to reach the tumor site. Surface modifications, mainly PEGylation, are widely used to reduce opsonization and, consequently, increase the circulation half-life of NPs [32,45]. If the nanocarriers can overcome the clearance processes, they must be transported to and accumulated at the tumor site.

### 2.2. Nanocarrier Transport by Enhanced Permeability and Retention-Based Passive Tumor Targeting

The enhanced permeability and retention (EPR) effect is the main mechanism described so far by which nanocarriers are more likely accumulated in solid tumor sites than in normal tissues [53,54,55]. The EPR effect exploits the proliferation of endothelial cells during rapid and defective angiogenesis (i.e., the formation of new blood vessels from existing ones) and the lack of adequate lymphatic drainage. There is a loss of tight junctions and significant gaps between endothelial cells [32] that varies from 100 nm to 2 µm, depending on the type and stage of a tumor [32,54]. The size of nanocarriers plays the most critical role in EPR-based passive tumor targeting. Only nanocarriers smaller than the gap can extravasate from the vasculature and accumulate in tumor sites [31,32,33]. The accumulated nanocarriers are retained due to the dysfunctional lymphatic drainage in tumors, which allows them to release drugs into the vicinity or inside the tumor cells [33]. In general, a higher retention of nanocarriers allows a higher extravasation and accumulation in the tumor sites, which is a better EPR-based passive tumor targeting. Nanocarriers with sizes between 30–200 nm are proposed as optimal for EPR-based passive tumor targeting [32]. Besides, EPR-based passive tumor targeting is time-dependent and requires a long circulation half-life of nanocarriers.

### 2.3. Nanocarrier Uptake

Upon nanocarrier–tumor cell contact after accumulation to the tumor site, the nanocarriers must cross the cell membrane to release the therapeutic agent and achieve the therapeutic function, often by endocytic or direct cellular uptake routes. The plasma membrane is folded into vesicles to engulf nanocarriers on the cell surface in the endocytic route (Figure 2). After endocytosis, nanocarriers are confined within intracellular vesicles, without direct or immediate access to the cytoplasm or cellular organelles [45,56]. Endocytosis pathways include clathrin-mediated endocytosis (also known as receptor-mediated endocytosis (RME)); caveolin-mediated endocytosis; clathrin- and caveolin-independent endocytosis; phagocytosis; and macropinocytosis [32,45,56]. The type of endocytosis is determined by numerous factors, including cell type, nanocarrier size, and receptor–nanocarrier interactions. Clathrin-mediated endocytosis is the preferred endocytosis pathway for nanospheres. Caveolin-mediated endocytosis can occur in negatively charged nanocarriers smaller than 60 nm with a nanorod shape. Large nanocarriers are endocytosed through phagocytosis and pinocytosis, while small nanocarriers are endocytosed through clathrin- or caveolae-mediated endocytosis. Then, nanocarriers must also escape from intracellular vesicles to achieve functional delivery. In nanocarriers’ direct cellular uptake route, nanocarriers cross the cell membrane by biochemical or physical processes, including direct translocation and lipid fusion, depending on the nanocarrier’s physicochemical properties [56]. In either of the two routes (i.e., endocytic and direct), the release of the therapeutic agent depends on the degradation/disassembly rate of the NPs and the NP–therapeutic agent interactions.

### 2.4. Active Tumor Targeting

Although EPR-based passive tumor targeting is the current basis for tumor therapy, it has limitations [32]. For instance, certain tumors do not exhibit the EPR effect, and small tumors or metastatic lesions do not exhibit intense angiogenesis. In tumors with the EPR effect, the permeability of vessels may not be homogeneous throughout the tumor, limiting the accumulation of NP-DDSs in the tumor site. Besides, if the nanocarriers reach the tumor site, there may be inefficient retention and unspecific cell uptake of non-targeted nanocarriers. In this context, a better understanding of tumors drives the development of nanocarriers to reach specific tumor sites and deliver anticancer therapeutic agents or carry out anticancer therapy in the tumor cells. 

Active targeting is the most promising mechanism to improve tumor localization efficiency. It involves molecular recognition, in which nanocarriers coated with ligands, such as antibodies, peptides, aptamers, and small molecules, bind selectively and strongly to a specific receptor on the cell surface [57]. Although nanocarrier surfaces can be functionalized with a high ligand density due to their high surface-area-to-volume ratio, the density needs to be carefully tuned to optimize the balance between diffusion, uniform distribution, penetration depth, and binding affinity [32]. Functionalization can be achieved by various conjugation methodologies recently reviewed by our group [58]. Conventionally, the ligands used for active targeting are specific to bind to overexpressed receptors in tumor cells compared to healthy cells (a mechanism known as active tumor targeting). In this way, active tumor targeting can increase the retention of nanocarriers in the tumor site.

Further, tumor-targeting nanocarriers enhance cell uptake by inducing receptor-mediated endocytosis in tumor cells, increasing the intracellular therapeutic agent concentration and cytotoxic effect [32,33]. Yet, they still need to rely on the EPR effect to pass through the vascular wall gaps and accumulate in the tumor site. Since not all tumors exhibit the EPR effect, active vascular targeting is a promising alternative for crossing the vascular walls and improving the tumor site’s nanocarrier accumulation. In contrast, tumor-targeting nanocarriers with non-mediated endocytosis ligands may sometimes be advantageous in solid tumors, in which the ligand–receptor binding affinity is so strong that nanocarrier tumor cell uptake is prevented [33]. In these cases, tumor cells kill through anticancer therapeutic agent release at the tumor environment. NP-DDSs have enormous potential for applications in cancer therapy as long as all (bio)chemical and physical requirements are considered.

## 3. Polymersomes as a Platform for Anticancer Therapeutic Delivery

Polymersomes are hollow spheres with bilayers enclosing an aqueous cavity assembled from amphiphilic copolymers. Polymersomes are analogous to liposomes and niosomes (small synthetic bilayer structures based on non-ionic surfactants instead of phospholipid or amphiphilic copolymers) [59]. According to the membrane bilayer, they can be unilamellar, multilamellar, and multivesicular. Unilamellar polymersomes of different sizes are mostly employed as DDSs, including small (<100 nm), large (100–1000 nm), and giant (>1 μm) varieties [15,28]. Unlike most NP-DDSs, polymersomes may encapsulate hydrophilic cargoes in their aqueous core and hydrophobic ones in the hydrophobic bilayer, making them versatile platforms to encapsulate polar and nonpolar cargoes and co-encapsulate two or more cargoes of different polarities [16]. 

Compared to their analogous liposomes and niosomes, polymersomes enjoy outstanding properties (e.g., chemical stability, mechanical and rheological enhanced properties, and permeability, among others) [15]. Unlike polymersomes in which amphiphilic copolymers’ molecular weights can be modulated and tailored, in liposomes and niosomes, the molecular weight or structure in lipids and non-ionic surfactants [60,61] are fixed; therefore, many properties cannot be modulated. In this sense, it is important to highlight that polymersomes have invaluable characteristics, mainly due to the chemical composition of the amphiphilic copolymer, synthesis method, and assembly conditions. Other properties, such as size, shape, surface features (functional groups, charges, and ligands), and functionalization process are also critical [17,62]. Polymersomes have demonstrated a higher stability towards oxidation and hydrolysis reactions and are more resistant to bending and stretching deformation during the body’s transport (resistance to a high shear rate of blood circulation and deformation through tiny vessels) or cellular processes (e.g., division and fusion, among others). Polymersomes also present lower later fluidity and a higher viscosity of the bilayer, which contributes to the low permeability of the encapsulated cargo from the inner polymersomes to the external site [15].

Encompassing the features mentioned above, adding the biological advantages through functionalization and composition, the multi-purpose abilities, and the possibility of conceiving polymersomes as small nanoreactors leads us to consider, as a wide range of literature supports, that these NP-DDSs are suitable platforms for facing the challenging environment in cancer and can advantageously compete with the current therapeutic strategies. Besides, within the polymeric alternatives in NP-DDSs, nanocapsules or nanospheres (solid particles) lack the biomimetic organization of an outer bilayer that can decrease their stability and cargo-retention efficiency [45]. Regarding vesicle-based architectures, lipidic analogous structures as liposomes present disadvantages, as mentioned before, making polymersomes contenders to liposomes when considering their comparable components [63].

Thus, given the importance of the chemical composition of amphiphilic copolymers, self-assembly methodologies, and functionalization processes in the development of polymersomes with tailorable physicochemical, pharmacological, and biological properties beneficial for NP-DDSs design, the following sections briefly describe (i) the types of amphiphilic copolymers and polymerization techniques, (ii) amphiphilic copolymer self-assembly and encapsulation methodologies, and (iii) some generalities about the characterization of polymersomes.

### 3.1. Amphiphilic Copolymer Types and Synthesis

Connecting immiscible blocks to a single junction point through covalent chemistry leads to amphiphilic copolymers, frequently originated by the polymerization of more than one monomer where one type of homopolymer is sequentially attached to blocks of the other one. The chemical nature of the segments remains different, enabling a dual or amphiphilic behavior in the presence of a solvent [64]. Depending on their architecture, the amphiphilic copolymers are classified as block, graft, random, alternate, dendronized, and gradient copolymers (Figure 3). Moreover, their topological features differ from one-dimensional or linear (block copolymers) to three-dimensional (branched or dendritic copolymers). Next, some definitions and general considerations related to the types of amphiphilic copolymers and their synthesis are given.

#### 3.1.1. Amphiphilic Block Copolymers

Block copolymers are hybrid macromolecules originated by linking together discrete linear chains incorporating dozens to hundreds of chemically identical repeating units, spontaneously assembled into exquisitely ordered soft materials, such as polymersomes [65]. The three main reported block copolymer architectures are AB, ABA, BAB, and ABC, considering blocks A and C to be hydrophilic and B as a hydrophobic segment. 

AB diblock copolymers are the simplest block copolymer structures, where a common junction point links two chemically different blocks [66]. ABA triblock copolymers are based on symmetric repetitive units containing three blocks of A and B monomers where the first and the third A blocks present the same molecular weight and chemical features and the middle block B has a different chemical nature [67]. BAB triblock copolymers are characterized because the hydrophilic segment is covalently connected to two outer B hydrophobic blocks [68]. In ABC triblock terpolymers, the amphiphilic block copolymers are based on hydrophobic and hydrophilic segments joined by covalent bonds in the same molecular structure [69].

The synthesis of the amphiphilic block copolymer is generally achieved by the sequential polymerization of different blocks, but occasionally the end groups of polymeric blocks are joined together in coupling reactions. Controlled/“living” polymerizations are the most frequent techniques employed to synthesize block copolymers where two or more monomers are polymerized sequentially and the polymer chains grow at approximately the same speed, resulting in a mixture of similar chain lengths [67]. Therefore, controlled/“living” polymerization techniques are principally employed in the synthesis of block copolymers, as the lack of chain termination in these reactions allows polymer blocks to be synthesized in stages by the sequential addition of different monomers [70]. In this manner, within controlled/“living” polymerization techniques, the ring-opening polymerization (ROP), reversible addition-fragmentation chain transfer polymerization (RAFT), and atom transfer reversible polymerization (ATRP) methods allow extensive control over parameters such as the composition and block lengths of the hydrophilic and hydrophobic chains [71]. A general description of each technique, including ROP [72,73], RAFT [70,74,75], ATRP [76], has been widely reported.

The polymersomes based on amphiphilic block copolymers are the most abundant and reported in the literature due to their repeated structure. Many examples of amphiphilic block copolymers involved in polymersomes generation have been reported. For example, one seminal investigation was reported employing asymmetric polystyrene--poly(acrylic acid) block copolymer (PS-*b*-PAA) synthesized by anionic polymerization to obtain various morphologies, including vesicles for which *N*,*N*-dimethylformamide-water mixture was used. This study proved that tuning the hydrophilic and hydrophobic ratios led to different morphologies [77]. Later studies aimed to highlight the advantageous features of polymersomes over phospholipid membranes. Discher and collaborators developed vesicles based on the neutral synthetic polymer poly(ethylene oxide)-block-poly(ethyl ethylene) PEE_37_-*b*-PEO_40_ (PEO is also referred to as poly(ethylene glycol) PEG). Those polymersomes were shown to be almost an order of magnitude tougher and sustained far greater areal strain before rupturing in comparison to unsaturated phospholipid liposomes during micropipette aspiration experiments [78]. Other reports improved amphiphilic block copolymers’ effectiveness for drug delivery through chemical modifications and auxiliary agents [79]. Recent studies have explored linear amphiphilic block copolymers due to their amenability for modification, monomer versatility, possibility to change blocks, molar mass, and great extent, valuable properties for biomedical applications and drug delivery [28]. For instance, Hou et al. developed a photoresponsive polymersome with potential cancer applications composed of an amphiphilic block copolymer, with pure poly(o-nitrobenzyl acrylate) (PNBA) as the hydrophobic block and poly(*N*,*N*′-dimethyl acrylamide) (PDMA) as the hydrophilic block, synthesized by RAFT polymerization. The polymersomes were self-assembled to co-encapsulated hydrophilic and hydrophobic cargoes [80]. 

In contrast to conventional block copolymers, Cao and coworkers reported the controlled formation of biodegradable fluorescent polymersomes from amphiphilic block copolymers composed of poly(ethylene glycol)-block-poly(caprolactone-gradient-trimethylene carbonate) (PEG-P(CLgTMC)) [81]. As seen, the hydrophobic block is a gradient copolymer in which the monomers’ composition of CL and TMC change gradually from one end of the hydrophobic block to the other. The designed block copolymers were synthesized via a modular polymerization approach. Many other investigations related to amphiphilic block copolymers have been reported, as evidenced in Section 5.

#### 3.1.2. Amphiphilic Graft Copolymers

Graft copolymers comprise branched molecular structures. In a central linear chain, the backbone is attached to polymeric grafts or side chains of different chemical natures that can be distributed randomly. In these structures, one or more homopolymer B segments are linked to backbone homopolymer A. The main difference between these configurations concerning block copolymers is that the last ones are based on two polymers connecting at each end without any branching point [82]. There exist three general synthesis methods for grafting copolymers including “Grafting to,” “Grafting from,” and “Grafting through” [83]. Graft copolymers are usually synthesized by free radical polymerization, cationic ring-opening polymerization (CROP), and anionic ring-opening polymerization (AROP) of a monomer in the presence of a preformed reactive polymer. The graft–backbone link can be accomplished via backbone radical or recombination reactions via monomer initiation.

Babinot and collaborators studied the amphiphilic graft copolymer poly(3-hydroxyoctanoate-co-3-hydroxyundecanoate)-*g*-poly(ethylene glycol) (PHOU-*g*-PEG) synthesized by thiol-ene addition. Polymersomes were prepared through the nanoprecipitation method in acetone to water [84]. Zheng et al. prepared graft copolymers containing methoxypoly(ethylene glycol) and ethyl-p-aminobenzoate side groups on a poly(dichlorophosphazene) backbone (PEG/EAB–PPPs) synthesized by ROP methodology and a substitution reaction. Then, they fabricated both polymersomes and encapsulated water-soluble anticancer drugs by a reversed emulsion and evaporation process [85]. Naolou and coworkers synthesized a poly(glycerol adipate)-*g*-(poly(3-caprolactone)-*b*-poly(ethylene oxide)) (PGA-*g*-(PCL-*b*-PEO)) by a ROP polymerization and a cycloaddition reaction. Worm-like aggregates of polymersomes were obtained through a nanoprecipitation method [86].

#### 3.1.3. Amphiphilic Random Copolymers

A random copolymer is one in which the monomeric units are located randomly in the polymeric chain, independent of the nature of the adjacent monomeric units [87,88]. Random copolymers have been less studied on the self-assembly of polymersomes than their block copolymer counterparts because their dispersity in molecular weight usually leads to polydisperse assembled structures [89]. Yet, the random copolymer can be synthesized more easily by a one-step radical polymerization of two or more different monomers or by modifying presynthesized or commercial (co)polymers. For example, Zhu et al. synthesized random copolymers from *N*-acroloyl-l-glutamic acid (NALGA) and *N*-acroloyl-dodecyl amine (NADA) units through radical polymerization to evaluate the self-assembly behavior of these random copolymers depending on the hydrophilic-lipophilic balance (HLB) [90]. Li et al. modified poly(acryloyl chloride) (PAC) with 2-(4-(4-ethoxyphenylazo)phenoxy)ethanol (EAPE) through a Schotten–Baumann reaction and the subsequent hydrolysis of unreacted acyl-chloride groups to obtain the hydrophilic carboxyl groups. Polymersomes were self-assembled from the obtained poly(2-[4-(phenylazo)phenoxy]ethyl acrylate-co-acrylic acid) (PPAPE) [91]. Kong’s group modified polysuccinimide with different moieties to obtain polymersomes with different properties and applications [92,93,94,95]. Tian [96], Dan [97], and Deshpande et al. [98] also reported polymersomes self-assembled from amphiphilic random copolymers. As mentioned, the variation in monomeric functionalities controlled the morphology and surface functionality of supramolecular assemblies. Further, the easy synthesis of amphiphilic random copolymers makes them promising amphiphilic copolymer alternatives for assembling polymersomes.

#### 3.1.4. Amphiphilic Alternate Copolymers

The alternate copolymers present an alternate monomeric unit structure within the polymeric backbones. Thus, the two comonomers copolymerize or arrange in a regular alternating sequence along the chain in these copolymers. In these systems, each propagating species prefers to add the monomer rather than react with its own type of monomer. Therefore, each A-unit is immediately incorporated into the polymer chain, followed by a B-unit and vice-versa [99,100]. Generally, the systems that can form alternating copolymers are electron-donor benzylidene monomers, such as styrene and stilbene, with electron-acceptor monomers, such as maleic anhydride and N-substituted maleimides [99]. 

The alternate amphiphilic copolymers can be prepared through free radical copolymerization [101]. However, amphiphilic alternating copolymers and their self-assembled structures have been less explored, being in an infant stage of development due to the difficulty in synthesizing the amphiphilic block copolymers [102]. Within the reports of such copolymer self-assembled structures, Wu et al. demonstrated the formation of polymer vesicles based on hydrophobic maleate esters and hydrophilic polyhydroxy vinyl ethers arranged in an alternating polymer. The unilamellar vesicles were obtained through horn sonication [103]. Chakraborty and coworkers designed a multi-stimuli temperature- and redox-responsive polymersome based on co-assembly of two π-amphiphiles containing an acceptor (A) (naphthalene diimide) and a donor (D) (pyrene) chromophore [104]. Different authors, such as Li [105] and Lv [106], among others [102], also obtained self-assembled polymersomes from alternating amphiphilic copolymers. Goswami et al. developed vesicles assembled in water from alternating copolymers, prepared through RAFT polymerization of methoxy poly(ethylene glycol) (mPEG) functionalized styrene (VBP) and fatty acid attached maleimide (MF) monomers P(MF-alt-VBP) [101].

#### 3.1.5. Amphiphilic Dendronized Copolymers

These polymers are macromolecules with dendritic wedges of different generations grafted throughout every repeated unit of the principal chain. Dendrimers are highly branched macromolecules that emanate from a central core, whose dendronized polymers can form various shapes attributed to the hindrance repulsion among dendritic structures [107] depending on the generation number and structure. Different conformations can be obtained by changing the generation number and, therefore, the size of the amphiphilic dendrimer head group. Due to the combination of hydrophilicity and branched conformation, the building blocks obtained can be used as the polar part or hydrophilic segment in amphiphilic copolymers.

The first study on amphiphilic dendrimer copolymer was based on polystyrene (PS) with poly(propylene imine) dendrimers with different generations -PS-dendr-(NH_2_)_8_- forming vesicular structures. The hydrophilic nature of poly(propylene imine) dendrimers could be employed as the amphiphile head groups, whereas the size and shape of the rest of the amphiphile could be altered [108]. Other examples of branching utility were proposed by del Barrio et al. [109] and then by Lin and collaborators [110] who employed the same dendritic structures. They developed azobenzene-containing linear-dendritic block copolymers with varied generation numbers obtaining various morphologies, including polymersomes. The copolymer was composed of PEG chains of different molecular weights as hydrophilic blocks and different generations of azobenzene-containing dendrons based on 2,2-bis(hydroxymethyl)propionic acid (bis-MPA) as hydrophobic blocks. The obtained polymersomes, therefore, were based on units of a solvophilic PEG-block linked to a solvophilic dendritic polyester that was attached to the solvophobic aliphatic block also linked to a solvophobic azobenzene block. It was found that polymersomes were formed with a long enough solvophobic aliphatic block length at a lower dendritic generation number [110], as reported by other authors, such as Chandrasiri [111] and Abad et al. [112].

#### 3.1.6. Amphiphilic Gradient Copolymers

Gradient copolymers are copolymers in which the monomers’ composition changes continuously from one end of the polymeric chain to the other [113,114,115,116]. The synthesis of gradient copolymers requires simultaneous initiation and uniform growth of all propagating chains involved in the polymerization process [116] and uses a polymerization technique that does not include termination reactions [114]. In this sense, the controlled/“living” polymerization techniques previously described (ROP, RAFT, and ATRP) are the most used for the synthesis of gradient copolymers [114,115,116]. Other controlled/“living” polymerization techniques, such as ring-opening metathesis polymerizations (ROMP), as well as nitroxide-mediated polymerization (NMP), can also be used [114,115,116]. The mechanism for ROMP [117,118] and MNP [119,120,121] have been briefly described.

Milonaki et al. synthesized a series of gradient copolymers via living cationic polymerization from hydrophilic 2-methyl-2-oxazoline and hydrophobic 2-phenyl-2-oxazoline (MPOx) [122]. Studies about the self-assembling behavior showed the formation of organized supramolecular nanostructures of different morphologies and structural characteristics depending on the copolymer composition. Specifically, the more hydrophobic gradient copolymer was self-assembled in well-defined vesicles [122]. In another example, Zhang’s group synthesized a gradient copolymer of acrylic acid and 2,2,2-Trifluoroethyl methacrylate P(AA-grad-TFEMA) via RAFT polymerization [123]. The fluorinated gradient copolymers were self-assembled in a three-dimensional co-flow focusing microfluidic device (3D CFMD), obtaining polymersomes depending on the copolymer concentration and capillary dimensions.

Although few experimental studies have investigated the assembly of polymersomes from gradient copolymers, a recent computational study demonstrated their potential for aggregating in different structures (i.e., worms, rings, and polymersomes) [124]. When the fraction of insoluble groups per chain is not high, spherical micelles of gradient copolymer can aggregate, forming multidomain structures despite a high interfacial tension. Conversely, when the fraction of insoluble groups is high enough, worms, rings, and polymersomes are practically monodomain. The researchers also found a new structure of multicompartment polymersomes. 

As described, there is a wide chemical and structural variability of amphiphilic copolymers used to assemble polymersomes. Remarkably, the nature of the synthetic monomer building blocks defines the vesicle degradability. Thus, polymersome self-assemblies may be degradable or non-degradable, depending on the selected monomers [125]. While degradable monomers include structures from aliphatic polyesters and polycarbonates, such as PEG [126], poly(lactic acid) (PLA) [127], polycaprolactone (PCL) [128], and poly(trimethylene carbonate) (PTMC) [129], non-degradable ones include conjugated or aromatic hydrocarbon blocks, such as poly(ethyl ethylene) (PEE) [78], poly(butadiene) (PBD) [130], and PS [131], among others [70]. Independent of the monomer degradability, the hydrophobic and hydrophilic amphiphilic organizations formed by their self-assembly constitute the bilayer of the polymersomes, defining their thickness, stability, and permeability. 

Examples of amphiphilic copolymers reported for polymersome self-assembly are summarized in Table 1.

### 3.2. Amphiphilic Copolymer Self-Assembly and Cargo Encapsulation

The assembly and organization of different amphiphilic copolymers mentioned above (block, graft, random, alternate, dendronized, and gradient) involve a balance of intermolecular forces, resulting in various morphologies, such as lamellae, double gyroid, cylinders, spheres, or vesicles [132]. This assembly also depends not only on the polymer type and methodology but also on the chemical nature of the cargo. Some molecular factors that influence the amphiphilic copolymer self-assembly include the number of blocks (n), the number of block types (k), the degree of polymerization of each block (Ni), and the associated binary segment–segment interaction parameters (Xij), where i and j correspond to chemically distinct repeat units. Other secondary factors include block flexibility, block length distribution (dispersity), and the sub-block structure, such as alternating, random, or tapered sequences of repeat units [65].

Regarding the computational field, simulations frequently assist and guide the experimental designs. Therefore, the prediction of particle behavior or morphology through Monte Carlo, molecular dynamics, and dissociative particle dynamics have contributed to the accurate simulation of polymer assemblies and their properties [133,134,135,136]. Thus, Discher and collaborators reviewed computational molecular dynamics schemes that lent insight into assembly and reported the physicochemical properties of different polymersomes and their effect on drug release and therapy [133]. Further, Ortiz and coworkers proposed a dissipative particle dynamic simulation on PEO-based block copolymer polymersomes for rupture simulations, providing valuable theoretic data for the subsequent experimental analysis [137].

Other simulation studies, such as the work of Chakraborty et al., described a molecular dynamic simulation revealing the shape change configurations and the effect of surrounding forces on PEO–PS diblock copolymer-polymersome performance [135]. Other studies, such as the one from Li et al. computationally designed a fabrication of Janus polymersomes based on the BC block copolymers at the interface between AB and AC hemispheres. Their fabrication was accomplished via a water-in-oil-in-water (W/O/W) double emulsion. The phase separation was studied through dissipative particle dynamics simulations [138]. Other computational and modeling studies have been proposed [139,140,141,142].

Experimentally, some parameters can predict the morphology or type of structure obtained through the hydrophobic and hydrophilic ratios and other parameters [70]. For amphiphilic block copolymers, the packing parameter (p = v/al), which considers the volume of the hydrophobic block (v), the contact area of the head group (a), and the length of the hydrophobic block (l), predicts the most likely conformation of the self-assembled structure in solution. p < 1/3 will suggests that spheres are formed, 1/3 < p < 1/2 correspond to cylinders, and 1/2 < p < 1 predicts the obtention of polymersomes [143,144]. An additional tool for predictable shapes is the parameter Fw, given by the hydrophilic segment’s molecular weight and the amphiphilic copolymer’s molecular weight ratio (Fw = Mn_hydrophilic block_/Mn_polymer_). Therefore, Fw > 0.5 corresponds to spherical micelles, 0.4 < Fw < 0.5 is related to cylindrical micelles, and 0.25 < Fw < 0.4 predicts polymersomes [15]. 

Furthermore, it is essential to note that the amphiphilic block copolymers are spatially organized in the bilayer depending on the type of amphiphilic block copolymer. In self-assembled structures from AB diblock copolymers, the hydrophobic B blocks are situated in the middle of the membrane, reducing the interfacial contact area with the aqueous solution in contact with the hydrophilic A blocks. With ABA triblock copolymers, polymersomes are built in a *trans* configuration, where cylindrical, hydrophobic interactions of B form the middle layer and the two A blocks form the exterior and interior layers ABA triblock copolymers can also be built in a U-shape formed by a hydrophobic loop creating the membrane’s middle layer, whereas the two hydrophilic blocks constitute the external and interior layers [145]. For BAB triblock copolymers, the two hydrophobic ends of the copolymer chain aggregate to form the central layer as in ABA, and the looped hydrophilic central block conforms to the exterior and interior layer. Finally, with ABC triblock terpolymers, the hydrophilic block with the more considerable block length forms the exterior layer of the assembly structure, whereas the other forms the interior layer [146].

Graft copolymers may conform spheres, cylinders and vesicles as AB copolymers and tend to form specially unimolecular micelles due to the association of backbone sequences [147]. For random copolymers, the HLB dictates the self-assembly [89]. The assembly of alternate amphiphilic copolymers depends on many parameters, including the HLB [101], solubility parameters, compatibility between block copolymers, and topology, among others [148]. The packing parameter can also predict the curvature and morphologies of dendronized copolymers [149]. However, dendronized copolymers are different in shape than conventional block copolymers, as the latter are linear, extendable chains lacking hyperbranched ramifications [108]. The proposed spatial organization in the bilayer of amphiphilic copolymers is schematized in Figure 3. The self-assembly methodologies can be classified into solvent-free and solvent displacement methods as follows [150].

#### 3.2.1. Solvent-Free Methods

In these methods, no organic solvents are present in the polymersome solution, as the amphiphiles copolymer are only hydrated in the aqueous medium. The film rehydration technique is the most common solvent-free method. Here, the amphiphilic copolymers are first dissolved in an organic solvent, then vacuum dried, leaving a thin layer of the amphiphilic copolymer on a solid surface and polymersomes are self-assembled with the subsequent addition of water. Another generally used solvent-free technique is the direct hydration method, where amphiphilic copolymers are directly hydrated from powder in an aqueous environment [151]. Polymerization-induced self-assembly (PISA) is another approach that has been used as a solvent-free polymer-specific methodology to form polymersomes. This method includes a hydrophilic micro-initiator or macro-chain transfer agent that polymerizes the hydrophobic block and appends the water-soluble segment. Morphologies progress from high to lower curvature structures, such as polymersomes, as the block length increases [15,152].

#### 3.2.2. Solvent Displacement Methods

Solvent displacement, also known as solvent switching, corresponds to those techniques where water-miscible organic solvent and an aqueous phase coexist, promoting the amphiphiles’ self-assembly before removing the organic phase. Thus, these methods involve water-miscible organic solvent (e.g., dimethyl sulfoxide (DMSO), acetone, dimethyl formamide (DMF), acetonitrile, tetrahydrofuran (THF), etc.) to dissolve the amphiphilic copolymers. Some examples are nanoprecipitation, direct injection, emulsion phase transfer, and microfluidics. In solvent displacement methods, the amphiphilic copolymer is dissolved in a water-miscible solvent, added dropwise into water under vigorous stirring, and, finally, the organic phase is removed by dialysis, freeze-drying, or evaporation [19,153]. The existing assembly methodologies can be adapted to encapsulate specific cargoes into polymersomes. Thus, different types of cargoes as therapeutic agents or dyes as imaging contrast agents and inorganic nanoparticles have been encapsulated within polymersome structures, leading to hybrid platforms for versatile applications in nanomedicine [2].

The two assembly processes are adapted to include the cargo in the suitable media, depending on their hydrophilic or hydrophobic nature. Therefore, whereas the structure’s core accommodates hydrophilic cargoes, the membrane or vesicle bilayer accomodates hydrophobic ones [154], and encapsulation can be during or post polymersome self-assembly. 

The during-self-assembly encapsulation can be top-down from bulk copolymers and bottom-up from unimers [155]. In the top-down approach, valid for the direct hydration assembly method, the polymeric film is submitted to shear rate, ultrasound, or stirring and water or buffer solutions are added to reach film hydration; meanwhile, polymeric chunks slowly detach, creating different phases from micelles to vesicles. Hydrophilic cargoes can be loaded by adding an aqueous solution containing the cargo to the polymeric film [156,157]. The hydrophobic cargoes are pre-mixed with the amphiphilic copolymer in an organic solvent to involve the polymeric film [155]. In the bottom-up approaches, such as the solvent switch methods, the amphiphilic copolymer previously dissolved in a water-miscible solvent is changed for a hydrophilic solvent [158]. Both hydrophobic and hydrophilic cargoes can be encapsulated into the polymersomes. Generally, the hydrophobic cargoes are incorporated in the organic phase with the amphiphilic copolymers and situated into the membrane after forming polymersomes. The hydrophilic cargoes are added to the aqueous phase in which the polymersomes are formed, located inside (core) the polymersome structure [159]. The solvent switch methods can be either a physical solvent exchange or a change in solvent conditions, such as pH or temperature, for both hydrophobic and hydrophilic cargoes. However, organic solvents often contribute to the degradation of sensitive molecules or are associated with toxicity and need to be removed from the solution generally via dialysis [160]. 

Cargoes can be encapsulated post-self-assembly into previously formed polymersomes. For example, electroporation, increasing membrane permeability by applying an external electric field, may lead to cargo uptake [161]. Cargoes can penetrate the self-assembled polymersomes by simple diffusion through the outermost permeable surface. The most representative self-assembly methodologies of polymersomes and encapsulation via the during-self-assembly process are schematized in Figure 4.

### 3.3. Surface Functionalization of Polymersomes

Encapsulation cannot be considered separated from release mechanisms and surface functionalization, as their synergy ensures the efficacy and efficiency of the therapeutic regime based on polymersomes as DDSs. Directionalization of polymersomes toward cancer cells or tissues is not only due to the natural occurring EPR effect but depends upon size- and surface-coverage-related properties, e.g., polymersome surrounding compatibility and surface characteristics recognizable by the target cells. Functional polymersomes may be mainly prepared by one of the following procedures: (i) conjugation of functional ligands to preformed polymersomes, (ii) self-assembly of end-group functionalized amphiphilic copolymers, (iii) use of polymers with biofunctional hydrophilic blocks or (iv) employing prodrugs. 

#### 3.3.1. Conjugation of Functional Ligands to Preformed Polymersomes

Conjugating ligands onto the surface of preformed polymersomes can be carried out through adsorption, entrapment, noncovalent or covalent coating, and layer-by-layer assembly. The noncovalent binding approach directly attaches multiple ligands onto the surface of polymersomes, including those based on affinity interactions, e.g., the biotin-streptavidin binding approach [162,163]. The covalent conjugation approaches increase the ligand-binding stability with higher site-specificity and reproducibility. To cite an example, click chemistry conjugation involves generating building blocks by adjoining small units with heteroatom links (C–X–C) in between [164] Other functionalization strategies employ different functional groups, such as thiol and vinyl sulfone [165], hydroxyl and amine [166], and aldehyde [167], present on the targeting ligand of the polymersome surface. For more details, please refer to our recent review on the functionalization of photosensitive nanocarriers [58].

#### 3.3.2. Self-Assembly of End-Group Functionalized Copolymers

These routes involve grafting the targeting ligand at the end group of an amphiphilic copolymer before forming polymersomes [164]. The copolymer functionalization may involve the chain end(s) modification with ligands, such as carbohydrates [168], proteins, or peptides [169], among others, before the self-assembly into polymersomes. One advantage of these methods is that the end-functionalized copolymer may be mixed with a non-functionalized polymer in a suitable ratio to control the surface density of the ligand. However, it is essential to highlight that the end-functionalization changes the packing parameter of the block copolymer and, thus, their self-assembly characteristics and behavior [164,170].

#### 3.3.3. Polymers with Biofunctional Hydrophilic Blocks

Amphiphilic copolymers can be synthesized to include biomolecule-containing hydrophilic building blocks. The hydrophilic blocks generally are glycopolymers [171], linear carbohydrate chains, and proteins or amino acids [172]. As these biomolecules act as hydrophilic blocks or side chain groups, a high degree of functionalization can be achieved at polymersomes’ inner/outer surfaces, enhancing cell interactions compared to end-group functionalized block copolymers [164].

#### 3.3.4. Prodrugs

Prodrugs can be considered as pharmacologically inactive chemical derivatives of a drug molecule requiring an enzymatic and/or chemical transformation inside the body to release the active drug [173]. Polymersomes seem to be suitable structures for the “drug-initiated” method in which a drug initiates the controlled polymerization of a monomer, leading to a drug-polymer prodrug. Therefore, a drug-moiety prodrug that may potentially enhance physicochemical and pharmacokinetic properties can also be linked to a polymer scaffold [174]. In this sense, amphiphilic copolymers may include different prodrugs into the polymer before assembling polymersomes, constituting a stimuli-responsive therapeutic agent [175,176].

### 3.4. Characterization of Polymersomes

The accurate and rigorous characterization of polymersomes is crucial for developing reproducible and trustworthy structures. The most extended techniques for polymersome characterization are mentioned below.

Light scattering methods correspond to a turbidity-based measurement that employs laser light scattering. The particle’s Tyndall effect (scattering) and the Brownian motion are colloidal phenomena associated with the dynamic light scattering (DLS) technique. The intensity of scattered light from colloidal suspensions depends on the scattering angle (*θ*) and the observation time and its modulation as a function of time corresponds to the measure of the particle’s hydrodynamic size. Therefore, the Brownian motion correlates with the particle’s hydrodynamic radius (R_h_). Static light scattering (SLS) is employed to obtain the shape-dependent radius of gyration (R_g_) among other parameters [177,178,179]. The ratio R_g_/R_h_ corresponds to the factor ρ, which predicts the shape of the dispersed particles. The nanoparticle tracking analysis (NTA) is also used to track and observe the motion of the particles and evaluate their light-triggered propelled motion, combining scattering with visualization [180,181,182]. Small-angle X-ray scattering (SAXS) is used to study the types of structure of colloidal size particles, providing information related to the polymersome bilayer and bilayer thickness [183]. Electrophoretic light scattering (ELS) is employed to assess the surface charge by measuring the electrophoretic mobility of polymersomes. This measurement is directly related to the zeta potential (ζ) parameter [184].

Microscopy methods use microscopes to measure precise information about polymersomes’ size, morphology, and composition. Electron-based microscopy techniques, such as scanning electron microscopy (SEM) and transmission electron microscopy (TEM), are the most employed techniques to elucidate polymersome size, morphology, lamellarity, and bilayer thickness. The most common sample preparation for TEM analysis is the negative staining of the sample solutions with heavy metal salts (e.g., uranyl acetate), aiming to better contrast their shape and polymersome layers in the image [185,186]. Cryogenic transmission electron microscopy (Cryo-TEM), based on cooling samples to cryogenic temperatures through quick-freezing (e.g., freeze-drying), has been widely extended to visualize polymersomes in their original shape. Cryo-TEM can overcome vacuum-induced sample deformation in TEM by employing a rapid cooling process, such as freezing-drying [151,187]. Fluorescence microscopy is involved in the emission of light that occurs within nanoseconds after the absorption of light, typically of a shorter wavelength. This technique allows the determination of the size, lamellarity, and concentration of a dye and can distinguish free and incorporated dyes. Thus, polymersomes can be visualized under fluorescence microscopy by employing fluorophores such as Nile red (NR), among many others [185,188,189]. Finally, atomic force microscopy (AFM) is another microscopy method for assessing mechanical, electrical, and surface properties. In situ AFM under a liquid droplet (liquid-AFM) allows visualization of assembled supramolecular structures in the wet state [190,191,192].

Electromagnetic methods, such as nuclear magnetic resonance (NMR) spectroscopy, determine atoms’ and molecules’ chemical and physical properties, describing the response of nuclei to an applied magnetic field [193]. The successful synthesis of the amphiphilic copolymers comprising polymersomes and the bilayer features, such as lamellarity and polarity, can be analyzed through NMR. The measurement of photocleavage mechanisms in both monomer and amphiphilic copolymers has also been studied employing NMR [194]. Electron paramagnetic resonance (EPR) constitutes another alternative involving a static magnetic field. The EPR technique is an analog-based technique to NMR in which electron spins are excited instead of the spins of atomic nuclei to study materials with unpaired electrons, which is useful for metal complexes or organic radicals [195]. The photodynamic activity of polymersomes, including photosensitizers (PSs), has been studied with the EPR technique, which detects the production of reactive oxygen species (ROS) with irradiation, indicating the formation of paramagnetic species [196]. 

Spectrophotometric and chromatography methods are valuable tools for confirming the amphiphilic copolymer’s structure and quantifying encapsulated cargo by selecting a specific wavelength [197] or a chromatographic column, solvent, or condition. Fourier transform infrared spectroscopy (FTIR) has been employed to corroborate the chemical structure of the amphiphilic copolymers and the presence of functional groups through the presence of their characteristic peaks [198,199]. UV-Vis spectroscopy and chromatography can, thus, be employed as a quantification strategy according to the cargo characteristics to quantify the cargo encapsulation efficiency (EE) and loading capacity (LC) [58]. Alternatively, UV-vis spectroscopy has straightforwardly monitored the photocleavage [80] and photoisomerization [200] properties in light-triggered polymersomes by evaluating the spectra evolutions upon light irradiation. 

## 4. Polymersomes for Stimuli-Responsive Anticancer Therapeutic Delivery

The physicochemical properties of the compartmentalized cell-like organization provided by polymersomes protect therapeutic agents and regulate cargo release. Hence, effective encapsulating, successful releasing, and specific functionalization are essential for cancer nano-therapy. However, those attributes are not enough for the current requirements, especially for variable, heterogeneous, and complex diseases, such as cancer.

As widely reported, polymersomes generally present an impermeable and robust membrane to slow down the release of engulfed and entrapped therapeutic agents. Membrane permeability is one of the most important properties that dictates their potential employment as cancer therapeutic nanocarriers, among other applications. Further, in living cells that exclude the diffusion of some water and certain gases, the hydrophobic structure within the polymersome membrane behaves as a barrier to solute transport. Thus, the passive release through diffusion is frequently delayed, leading to the requirement of a boosting mechanism to improve the disruption, poration, or degradation of the polymer membrane [16,201]. Therefore, membrane permeability is imperative and may be modulated by designing polymersomes with intrinsic permeability, the formation of selective or biohybrid polymersomes, and the introduction of chemistry moieties. For the first strategy, the mechanical stability of polymersomes is generally related to their membrane thickness, also associated with the building units’ molecular weight. It has been reported that the higher molecular weight of polymers leads to membranes that are several nanometers thicker and may decrease membrane fluidity. In this sense, molecular weight, especially of hydrophobic segments, dictates the aggregate dimension regarding the hydrophilic–lipophilic balance (HBL) or polymer chain length, permeability, and membrane thickness. Therefore, the diffusion processes are restricted with increasing molecular weight due to the higher chain entanglements [202,203,204,205]. Various studies involving membrane tunability have been reported [166,206,207]. 

A different strategy to modulate membrane permeability implies the introduction of membrane proteins, transporters, pore-forming peptides, or ion channels. The introduction of those heterogeneities allows pore selectivity, providing them adaptable features, such as the exchange of ions or small organic molecules [201]. The permeability required for reactions or molecule transport has also been proposed via the inclusion or integration of channels into the polymeric membrane. It has been employed with polymersomes as nanoreactors (and/or nanocontainers) comprising supramolecular assemblies. The polymersomes contain and protect catalysts and various substances, such as enzymes, from the environment, simultaneously mediating the passing of molecules through the membrane to the inside of the polymersome cavity where chemical reactions can be accomplished [208,209,210,211]. However, many of the membrane proteins and biomolecules are structurally fragile, and few of them lack the robustness to endure reconstitution within synthetic vesicles. Therefore, for incorporating those biological structures that need high stability, such as proteins, adapted surfactants and determined pH conditions are required. Moreover, the amphiphilic copolymer and biomolecule mixture before assembly should avoid employing organic solvents, which are often required in amphiphilic polymer dilution, as they could perturb and denature biomolecules [201,205,212].

Incorporating chemical groups that can be stimuli-controllable-reactive is imperative for the third approach to tuning polymersome membrane permeability. Hence, one crucial characteristic of nanocarriers, especially polymersomes for therapeutic delivery, is related to the “smart” performance according to their accurate release in a specific place and time. The “smart” performance is called here “stimuli-responsiveness,” involving valuable strategies for the modulation of nano-construction activity in cancer nanomedicines [52]. The general approaches involved in stimuli-responsive polymersomes are the external or exogenous and the internal or endogenous stimuli. The activated response leads to the nonreversible disassembly of polymersomes or a reversible disruption of the system permeability. External stimuli are related to physical changes (i.e., temperature, ultrasound, electric, magnetic field, or light), whereas internal stimuli are usually linked to chemical or biological variations (i.e., pH, milieu, electrochemical changes, or enzyme production) [213,214,215,216,217,218]. 

An interesting type of stimuli-responsive system is that with multiple simultaneous stimulus responses. This system can present an external stimulus response, such as temperature or light, to control the cargo delivery. In contrast, another external stimulus or an internal one, such as acidic pH, high redox potential, elevated concentration of ROS, or high temperature in the tumoral tissues can synergistically activate and guide the nanostructures to the specific therapeutic zone [52,219,220,221,222]. Hence, considering the complexity and multi-step procedures involved in cancer therapy, “take out two targets with one shot” strategies would effectively trigger drugs with superior performance to single therapeutic strategies [223]. Moreover, dual-response and multi-responsive polymeric vesicles combine the effects of various stimuli, improving the therapeutic systems’ performances.

Within the redox and enzymatic (e.g., glutathione) stimuli [13,224,225], the most explored stimulus to date is pH, which is often used to trigger polymers and polymersomes for cancer therapy [215,226,227,228,229,230]. The cancer microenvironment presents typical acidification of the extracellular milieu (low pHe) and simultaneous intracellular alkalinization in the cytoplasm (high pHi). In normal cells, the pHi and pHe are approximately 7.2 and 7.4, respectively. Consequently, whereas normal tissues have a higher extracellular pH than intracellular pH, cancer is precisely the opposite [231,232,233]. Thus, pH sensitivity in polymeric nanocarriers is given by incorporating protonable groups, labile acidic bonds in the polymers, or the pH-responsive “PEG detachment” achieved extra- or intracellularly [227]. Hence, the pH in single and multi-response systems has been widely studied for pH-responsive polymersomes in the last few years. However, internal stimuli, such as the redox milieu, enzymes, and pH, represent internally activated sources that vary from patient to patient or among organisms. Mainly, the acidic pH in perivascular regions is usually far from the blood flow, leading to a lack of nanocarrier response. Moreover, it has been reported that pH variations may not significantly differ in healthy and tumor tissues [234]. 

Electromagnetic irradiation has promoted chemical conversions and catalyzed fundamental processes, even at the beginning of life. Photosynthesis, circadian rhythm, and the photoisomerization of the visual pigments in our eyes when we see, along with other complex mechanisms in nature, have evidenced the relevance of light. Light is an excellent and attractive option as an external stimulus, owing to its non-invasive nature and minimal tissue damage, depending on wavelengths and intensities. Moreover, light presents an excellent remote and spatial-temporal control over the release of therapeutic agents from structures as polymersomes in direct or indirect ways to induce phototherapies [29,235,236]. Therefore, electromagnetic irradiation in single- or multi-responsive polymersomes comprises a versatile, relevant, and affordable clinically significant stimulus for boosting photo-mediated processes, which leads to tunable therapeutic agent release. Despite, the electromagnetic spectrum including irradiation from longest to shortest wavelength, i.e., radio, microwave (MW), NIR, visible, UV, X-rays, and gamma-rays, only UV irradiation (10 nm < λ < 400 nm) and NIR irradiation (760 nm < λ < 1500 nm) are currently used as a stimulus for cargo release from DDSs. UV irradiation has shallow tissue penetration depths (< 0.1 cm) due to its absorption by the skin, blood, and tissues [235]. However, UV wavelength irradiation is restricted due to phototoxicity, although it can be used for therapeutic agent release for topical treatment of the skin and mucosa [235,237]. In contrast, NIR irradiation displays negligible phototoxicity and can penetrate more deeply into biological tissues (from 0.1 cm up to 10 cm). NIR-enhanced penetration capacity is related to hemoglobin, water, and lipids, which have low absorption in the NIR region [235,236,237,238]. Visible irradiation (380 nm < λ < 760 nm) is generally employed to reverse the light response caused by UV irradiation [239]. 

## 5. Light-Responsive Polymersomes for Anticancer Therapeutic Delivery

As previously mentioned, light-responsive polymersomes are attractive, especially for their non-invasive nature and remote and spatial-temporal control over the release of therapeutic agents. Light-responsive polymersomes are typically self-assembled from an amphiphilic copolymer in the presence of light-activable moieties, including functional chromophores, PSs, photothermal conversion agents (PCAs), and/or light-responsive NPs [240]. Some light-activable moieties are represented in Figure 5. These light-activable moieties can be covalently attached to the amphiphilic copolymer or encapsulated in any polymersome compartments. Either of them first absorbs the light and then is converted to a chemical signal through photoreactions, photosensitization-induced oxidation, heat, and photoconversion. These reactions induce the disassembly and disruption of polymersomes and promote cargo release. 

We summarize the recent development in light-responsive polymersomes as DDSs with on-demand anticancer therapy. As depicted in Figure 6, there are five main therapeutic agent release mechanisms of the light-responsive polymersomes, including (i) the photoreaction of chromophores, (ii) the photothermal effect, (iii) photo-oxidation, (iv) the upconversion energy process, and (v) multifunctional release. These mechanisms are briefly described below within the wide range of possibilities of light-activable moieties (Figure 5). At last, the representative light-responsive polymersomes using multifunctional DDSs are summarized.

### 5.1. Controlled Release Induced by Photoreaction of Chromophores

In the most common mechanism of light-induced release of cargoes, polymersomes are self-assembled from amphiphilic copolymers with a functional chromophore moiety that goes from the ground state (G) to the excited state (E) through light absorption. Then, it undergoes a photochemical reaction [236], i.e., photoisomerization, photorearrangement, and photocleavage. Conventionally, UV photoreaction is induced by one-photon absorption, while NIR photoreactions are induced by two-photon absorption. 

Photoisomerization involves photochromism i.e., the reversible change of a specific chemical group (photoswitch) between two forms by the absorption of electromagnetic irradiation of a chromophore moiety into its structural isomers [239,241]. Photoisomerization is processed via one-photon UV irradiation and is reversible under visible irradiation [239]. Photochromic chromophores include azobenzene (AZO), spiropyran (SP), dithienylethene (DTE), and their derivatives (Figure 5). AZO moieties isomerize from its linear *trans* form to the bent *cis* when irradiated with UV (λ = 300–400 nm) while isomerizing back by visible light irradiation (λ = 425–500 nm) [239,242,243]. SP and its derivatives isomerize from the hydrophobic SP state (also called the closed form) to the hydrophilic zwitterionic merocyanine (ME) state (also called the open form) under UV irradiation at 365 nm, while the reverse process is triggered by visible light (620 nm) [238,239]. DTE have generally aromatic groups bonded to each end of a carbon–carbon double bond. This specific structure allows the photoisomerization from the *trans* isomer to the *cis* isomer and changes between ring-open and ring-closed photo isomers [239].

In a photorearrangement reaction, the hydrophobic segment of an amphiphilic copolymer converts into a hydrophilic segment under light irradiation. This conversion alters the polarity of the polymersomes and destabilizes them to release the cargo. As a type of photorearrangement moiety, hydrophobic 2-diazo-1,2-naphthoquinone (DNQ) changed into a hydrophilic 3-indene carboxylic acid (3-IC) (Figure 5) through the NIR- or UV-induced Wolff rearrangement reaction [238,239]. 

The photocleavable group (also known as photoremovable or photolabile) corresponds to molecules that can be irreversibly photoactivated, removing the photolabile group. The design of efficient cleavable molecules requires good leaving groups in their structure, and the products formed after irradiation should be stable to avoid any recombination while increasing the cleavage efficiency. Photolabile groups, such as the ortho-nitrobenzyl (ONB) group, with all its derivatives, have been widely employed as photolabile linkers [244,245,246]. Other components, such as coumarin and pyrenylmethyl ester, can also participate in the photocleavage reaction (Figure 5). The photocleavage process of ONB and coumarin can be triggered via either one-photon UV light and/or two-photon NIR light, while the photocleavage of pyrenylmethyl ester is promoted by UV light irradiation. Table 2 summarizes the polymersomes designed for the controlled release induced by the photoreaction of chromophores.

Song and collaborators reported plasmonic vesicles employing AuNPs (14 nm) coated with polymer brushes of hydrophilic PEG and hydrophobic PNBA, containing the photolabile ortho-nitrobenzyl ester moiety as building blocks. The coated AuNPs were synthesized through reactions via ATRP and resulted in amphiphilic AuNPs (Au@PEG/PNBA). In the “grafting to” step, the PEG grafts enabled the conjugation with folate as the targeting ligand to folate receptors is overexpressed in many types of human cancer cells. The Au@PEG/PNBA were assembled through the film rehydration method, which led to the obtention of particles with a hydrodynamic diameter of 210 nm. The ONB ester moiety contained in the hydrophobic PNBA was cleaved upon UV irradiation (365 nm) and changed the PNBA chains into hydrophilic poly(acrylic acid) (PAA), leading to the disassembly of the vesicle, evidenced through the size decrease. In vitro experiments with MDA-MB-435 breast cancer cells evidenced that in 40 min of incubation, the folate-conjugated particles bonded to the MDA-MB-435 cells were taken up through folate receptors. Additionally, the hydrophilic doxorubicin (DOX) drug was loaded into the vesicles through a pH-gradient method during the film rehydration. The results suggested that the loading content of DOX was stabilized when increasing the DOX concentration up to 30% when the weight ratio of DOX and vesicles reached 50%. Further, after 15 min of photoirradiation, the Au@PEG/PNBA reached an 80% release of the loaded drug, and the cytotoxicity test showed an enhanced cytotoxic effect when the particles were irradiated and released the subsequent drug. Besides, the half-maximal inhibitory concentration of the folate-targeted vesicles against the MDA-MB-435 cells was 0.44 µg mL^−1^, 5-fold lower than the non-targeted vesicles, confirming the therapeutic potential of the vesicles [247].

Liu’ group proposed a simultaneous approach to crosslinking polymeric vesicles and permeabilizing the originally impermeable bilayer membranes of vesicles by taking advantage of light-actuated traceless cross-linking reactions within the bilayer [248]. They employed a light-triggered crosslinking strategy with an amphiphilic block copolymer containing photolabile carbamate-caged primary amine moieties. In this first work [248], polymeric vesicles were self-assembled by nanoprecipitation of PEO_45_-*b*-PNBOC_30_, where photolabile carbamate-caged primary amine moieties were located within the hydrophobic membrane bilayers of polymeric vesicles. Upon UV irradiation at 365 nm, the PNBOC block was transformed into a PAEMA block because photocleavage of the ONB functionalities generated ortho-nitrosobenzaldehyde and primary amine moieties. Then, extensive interchain/intrachain amidation reactions occurred due to suppressed amine pKa within the hydrophobic membrane. During cross-linking, the bilayer hydrophobicity-to-hydrophilicity transition and membrane permeabilization with retention of morphology occurred (Figure 7). It was demonstrated that the light triggered co-release of both hydrophilic DOX and hydrophobic NR.

From the same group, Wang et al. proposed photochromic polymersomes with photo-switchable and reversible bilayers resulting from the assembly of the amphiphilic PEO-*b*-PSPA diblock copolymers synthesized by controlled radical polymerization. The diblock copolymers contained the SP moiety in the SP-based methacrylate monomer containing a single ester linkage (SPMA). This SP moiety would be able to reversibly isomerize between the hydrophobic colorless ring-closed state and the ring-opened MC when irradiated with λ1 < 420 nm and λ2 > 450 nm. Moreover, when irradiated with λ1, the SP–MC transition led to the MC hydrophilic zwitterionic derivatives, which also promoted the obtention of stable polymersomes through non-covalent interactions (hydrophobic interactions, hydrogen bonding, and π-π stacking). The assembled structures were obtained via the nanoprecipitation method, resulting in 450 nm size polymersomes. The subsequent λ2 propelled a switched-off permeability with the reversible photo-triggered SP-to-MC transitions, which led from the non-permeable to the permeable state in the membrane towards small molecules. The light-mediated release of the hydrophilic anticancer drug 2′-deoxy-5-fluorouridine (5-dFu) was evaluated within 8 h, resulting in the releasing of over 90% and <10% from the 365 and 530 nm irradiated polymersomes, respectively, which drove the MC–SP transition within the membrane. The authors confirmed a controlled release profile with the small 5-dFu molecule (246 Da) contrary to larger molecules, such as DOX (580 Da; positively charged) or calcein dye (623 Da, negatively charged) [249].

Song and collaborators developed UV-responsive photocleavage polypeptide-glycosylated poly-(amidoamine) (PAMAM) dendron amphiphiles (PGDAs), obtained by copper(I)-catalyzed azide-alkyne cycloaddition chemistry. It enabled the assembly of sugar-coated polypeptide vesicles (polymersomes) while delivering ovalbumin (OVA) antigen into mammalian cells [198], which could escape from endolysosomes into macrophage cytoplasm, activating the cellular immune response. Thus, glycosylated polypeptides presented carbohydrate residues located in the terminal ends or side chains, useful for cell recognition, adhesion, and binding, among others, through structures such as lectins, including concanavalin A. Hence, UV-light activation transformed the polymersomes into micellar aggregates and led to the complete micellar disassembly at pH 7.4 and 5, respectively. The encapsulation of OVA-FTIC was accomplished with an LC and LE of 10.2 and 46.4 wt. %, respectively. Moreover, the OVA-FTIC-loaded polymersomes presented a 121 ± 8 nm hydrodynamic diameter, a membrane thickness of 21.4 ± 1.4 nm, and ζ potential of −18.1 mV due to the negatively charged OVA-FITC. Further, the cell uptake was evaluated with RAW264.7 macrophages via an endolysosome pathway, enhancing the TNF-α level upon light irradiation and stimulating the immune response. Then, the sugar-coated photo-triggered polymersomes showed potential for cancer immunotherapy. 

Hou and coworkers proposed the synthesis of amphiphilic block copolymers with PDMA as the hydrophilic block and PNBA as the hydrophobic block (PDMA-*b*-PNBA) and using bulk RAFT with 2-(Dodecylthiocarbonothioylthio)-2-methylpropionic acid (PDMA) macro-RAFT agent [80]. DOX and NR dye were co-loaded into polymersomes to evaluate the photo-triggered releases of both drugs. Polymersomes were self-assembled by the emulsion method. After 15 min of UV light irradiation at 365 nm, the polymersomes demonstrated a clear photoresponse. The hydrophobic block was transformed into a hydrophilic one in the photocleavage reaction of the ONB groups, thereby dissociating the polymersomes with the simultaneous co-release of both DOX and NR. The release of DOX from the polymersomes depended on the irradiation time.

Zhou et al. reported light-responsive polymersomes self-assembled from no conventional amphiphilic polymer described so far [250]. Researchers synthesized an amphiphilic monomer (C12NB) in which photolabile an ortho-nitrobenzyl moiety connected two hydrophobic alkyl chains and a hydrophilic ammonium group. Cationic ammonium salt provided the hydrophilic segment and two dodecyl-substituted ONB derivatives provided hydrophobic and light-responsive functions. The lipid-like amphiphilic polymer PC12NB was obtained by free radical polymerization of C12NB. PC12NB was self-assembled into polymersomes by nanoprecipitation with the simultaneous encapsulation of DOX. After self-assembly, quaternary ammonium enabled electrostatic adsorption of folic acid (FA) onto the surface of the polymersomes, thus achieving the targeting of cancer cells with an overexpression of the folate receptor. The photocleavage reaction at 365 nm of the ONB moiety disintegrated polymersomes by changing the polymer structure from a cationic amphiphilic state to a zwitterionic hydrophilic state, thus realizing photo-triggered drug release. Cytotoxicity studies showed that DOX-loaded polymersomes functionalized with FA had high cytotoxicity to HeLa cells after photo-triggered release. 

Soo Kim and coworkers proposed a photo-reactive oligodeoxynucleotide (PRO)-embedded vesicular polyion complex (PIC) (PROsome) loading the nuclear-enriched abundant transcript 2 (NEAT2)-targeting ASO (asNEAT2) as the therapeutic agent for intracellular gene knockdown. The structures were formed in an aqueous solution with the mixture of PROs as an anionic reversible crosslinker, asNEAT2, and the cationic PEG-block polypeptide PEG-block-poly[*N*-(5-aminopentyl)-α,β-aspartamide] (PEG-P(Asp-AP)) as a cationic component. The PROsome presented a UV365-triggered crosslinking for the obtention of crosslinked structures ((X-)PROsomes), and UV312 triggered de-crosslinking and led to the payload release. The PROsome was assembled through the mixture of PEG–P(Asp-AP):PRO:asNEAT2 (molar charge ratios 2:1:1) followed by UV365 irradiation for the crosslinking between the photo-reactive vinyl groups ^CNV^K-modified D-threoninol (^CNV^D) in the PRO and the T-containing oligodeoxynucleotides in the asNEAT2, forming a cyclobutene structure for conferring stability to the structures in the biological milieu. The vesicular PIC structures presented a hydrodynamic diameter of 80 nm with a PDI around 0.1. Then, the formed cyclobutane could be de-crosslinked with UV312 irradiation. Hence, the results demonstrated the photo-switchable capacity of PROsomes for ASO release. Moreover, the performance of the structures as carriers in a biological model was studied in human lung cancer cell culture (A549 cells). The gene knockdown efficiency of X-PROsomes against NEAT2 levels (highly expressed in human cancer, including lung cancer) was evaluated after 0.5 min of UV312 irradiation, reaching up to 80% of efficiency for 48 h post-incubation, which probed the capacity of the vesicles for the photo-triggered enhanced gene knockdown [251].

### 5.2. Controlled Release Induced by the Photothermal Effect

In addition to the controlled release induced by the photoreaction of chromophores, polymersomes can use light to control the release induced by the photothermal effect of photo-oxidation, i.e., light is not used to directly promote the disassembly and disruption of polymersomes. For controlled release induced by photothermal effect, polymersomes contained PCAs, which include inorganic materials, such as gold nanostructures (nanorods -AuNRs-, nanospheres, nanocages, and nanoframes), Prussian blue, metal chalcogenides, carbon-based nanomaterials, as well as organic molecules, such as polyaniline, boron-dipyrromethene (BODIPY), and indocyanine green (ICG) (Figure 5), among others [235]. In general terms, PCAs are excited by a specific wavelength of light and convert the photon energy into localized heat, accelerating the diffusion of cargo and disassembling and disrupting thermosensitive polymersomes to release the cargo. The heat produced in this process allows cargo release and is the principle of photothermal therapy (PTT).

PTT comprises the activation of PCAs by laser pulses with electromagnetic irradiation, such as radio, MW, NIR, or visible light, generating heat for the thermal ablation of tissues such as cancer tumors. Indeed, it has been reported that cancer cells have poor endurance to heat [252]. The photothermal effect of organic materials is associated with the transition of electrons in molecules when they are irradiated with light subjected to absorption, scattering, and transmission [253]. Additionally, carbon and inorganic materials, such as metals and metal oxide-based NPs, also offer advantageous photothermal features, as they present high photothermal efficiencies and intrinsic optical properties, such as the localized surface plasmon resonance (LSPR), among others. Generally, the photothermal mechanism involves the PCAs’ energy absorption, which is promoted from the G singlet state to an E singlet state. The PCA then experiences nonradiative vibrational relaxation, returning to the G state through a collision among the excited PCAs and the surrounding molecules. Thus, the kinetic energy heats the surrounding environment. In this context, the activated PCAs, as “enhancers” to heat-targeted tissues, are based on PCAs that absorb light of a particular wavelength and transform it into valuable energy. In the case of PTT, they transform it into heat, increasing the temperature of the milieu, which is also called hyperthermia. In clinical settings, hyperthermia refers to treating diseases, such as cancer, through heating, inducing arduous processes, such as lysis of cell membranes, denaturation of proteins, or even evaporation of cytosol, leading to induced cell death [252,253,254,255,256]. Examples of polymersomes used in PTT are summarized in Table 3.

Song et al. developed plasmonic vesicles with AuNRs (13 nm × 51 nm) coated with brush amphiphilic copolymers, synthesized via sequentially conducted “grafting to” and “grafting from” reactions. Herein, the hydrophilic PEG and the ATRP initiator 2,20-dithiobis [1-(2-bromo-2-methylpropionyloxy)]ethane were simultaneously attached to gold nanocrystals through covalent Au-S bonds as a ligand exchange. Then, the nanocrystals were used as macroinitiators for the ATRP from hydrophobic monomers. Thus, the authors examined the assembly of AuNRs with PEG and the hydrophobic poly(methyl methacrylate) (PMMA) grafts (AuNR@PEG/PMMA). The plasmonic vesicles were assembled by a film rehydration method obtaining particles of 200 nm. Further, due to the photothermal effect of AuNRs, the authors evaluated the light-induced deconstruction of the vesicles via 785 nm irradiation. Thus, the results evidenced a collapse and subsequent transition to non-spherical morphologies after 2 min of irradiation, confirming the light-responsive vesicle performance, followed by a thermal mechanism able to disrupt the cavity for potential release of the active component [257].

Sun et al. developed an AZO-containing vesicle assembled from the copolymer poly(*N*-isopropyl acrylamide)-block-poly{6-[4-(4-methoxyphenylazo)phenoxy] hexylacrylate} (PNIPAM-*b*-PAZO). The assembly method was nanoprecipitation, where the simultaneous loading of the hydrophobic Fe_3_O_4_ magnetic NPs (MNs) (20 nm) was accomplished with a widely reported photothermal effect beneficial for hyperthermal cancer treatment and a 5 µm vesicle diameter. Further, the authors evaluated the photothermal effect through irradiating MN-doped vesicles with different light intensities. Herein, it was evidenced that the vesicles tended to aggregate around the NIR (60 mW µm^−2^) laser-focused vesicle (target vesicle), and neighboring vesicles fused to the target vesicle, leading to an increased vesicle size. Then, as the NIR irradiation power increased (90 mW µm^−2^), a fission phenomenon followed the fusion process. Moreover, when the light density was 120 mW µm^−2^, the target vesicle burst in 1 s, and neighbor vesicles migrated to the light spot and burst. These fusion and fission mechanisms were related to the conversion of NIR light into heat (photothermal effect) in the vesicle shell containing the MNs, which finally destabilized the vesicle. However, when the MN concentration was kept at 0.01 mg mL^−1^ and UV irradiation (365 nm) was employed to irradiate the whole sample, unlike NIR (focused on the target vesicle), the authors suggested that the photoisomerization from the azobenzene moiety could contribute to the vesicle fission with the photothermal effect as a complete releasing mechanism [258].

Further, the plasmonic nanoparticles have been suggested as photothermal agents through the excitation of their LSPR. For example, DiSalvo et al. developed micro and nano-polymersomes employing the diblock copolymer PBD_35_-*b*-PEO_20_ through the gel-assisted rehydration method [189]. Polymersomes containing 2.5 nm hydrophobic dodecanethiol plasmonic AuNPs within the hydrophobic membrane promoted poration and/or rupture by a 532 nm wavelength pulsed laser. The results showed that the releasing of fluorescent dye fluorescein isothiocyanate-dextran (FITC-dextran) within the aqueous particles’ lumen could be adjusted by modulating the single-pulse energy periods during a timescale from seconds to minutes through pore formation and also by varying the AuNP concentration. Thus, the authors explained the mechanism of polymersomes rupturing with nanobubbles formation around excited AuNPs upon thermal relaxation leading to the loss of membrane stability.

### 5.3. Controlled Release Induced by Photo-Oxidation

Like the controlled release induced by the photothermal effect, the controlled release induced by photo-oxidation uses light indirectly to promote the disassembly and disruption of polymersomes. In this case, polymersomes contained PSs. Examples of PSs are inorganic materials (zinc oxide (ZnO) and titanium oxide (TiO_2_)), carbon-based materials, and organic materials (phthalocyanine derivatives, cyanine dyes (Cy), and chorin e6 (Ce6)) (Figure 5) [235]. Under light irradiation, PSs generate ROS, including singlet oxygen (^1^O_2_), super oxide anion (O_2_**^•^**^−^), hydroxyl radical (OH·), and hydroxyl anion radical (HO^•−^). ROS production is the basis of photodynamic therapy (PDT).

PDT is a non-invasive treatment for various malignancies, such as cancer. The molecular mechanisms of PDT are mainly based on three non-toxic components that are able to produce the desired therapeutic effects in tissues through a mutual interaction between the PS, light in a particular wavelength, and oxygen dissolved in the cells [259]. The PS absorbs light energy and transfers it to non-absorbing molecules. The energy associated with the electron transfer produces highly ROS, which promote the oxidation of biomolecules, causing cell death and tissue necrosis. There are two principal mechanisms of the photodynamic reactions depending on oxygen inside cells. In both of them, the first stage may occur inside the cell when the PS is irradiated with a light wavelength coinciding with the PS absorption spectrum. It is converted from the singlet basic energy state S0 into the excited singlet S1 due to the photon absorption, and one part of the energy is radiated as quantum fluorescence. The remaining energy directs a PS to the excited triplet state T1, which is the therapeutic form of the compound. Then, for the type I photodynamic reactions, the triplet state T1, the PS can transfer energy to the biomolecules from its surrounding environment. Thus, electrons or hydrogen interact with oxygen molecules forming ROS that can destroy biomolecules. In the case of type II reactions, the PS transition into the excited triplet state enables the energy to be transferred directly to the oxygen in the energetic base state (the basic triplet state). Hence, the PDT destroys cancerous cells and vascular damage through photochemical and subsequent oxidation reactions, where the damage caused by the free radicals and ROS presents a short lifetime (10–320 ns) [259,260,261,262,263,264]. Table 4 summarizes the polymersomes designed for the controlled release induced by photo-oxidation.

Hsu et al. proposed porphyrin-embedded polymeric vesicles based on a four-armed porphyrin–polylactide (PPLA) conjugate. The conjugates were composed of four-armed PPLA conjugates via ROP with meso-tetra-(p-hydroxy methylphenyl) porphyrin as an initiator and calcium di-2-[(2-dimethylamino-ethylimino) methyl] phenol as a catalyst with the porphyrin as a PS. The PPLA porphysomes were assembled via the oil-in-water solvent evaporation method with an average hydrodynamic diameter of 52 nm and high stability after 1 month of storage. Moreover, the shape factor of the PPLA porphysomes was similar to the typical value 1 corresponding to hollow spheres, and the authors evaluated the generation of ^1^O_2_ in an aqueous solution without disruption. Additionally, the uptake of PPLA porphysomes by HeLa cells was confirmed, as characteristic fluoresce was evidenced in the cytosol. Further, the cytotoxicity analysis evidenced a decrease in the cell viability to 21% from a light dose increment from 1.4 J cm^−2^ to 2.1 J cm^−2^ with a porphyrin concentration of 2.8 µM, confirming the PDT potential of the proposed structures [265].

Baumann et al. developed nanoreactors for encapsulating PSs and protein conjugates inside a polymeric vesicle as ROS generators. The amphiphilic triblock copolymer was poly(2-methyl oxazolyne)-poly(dimethyl siloxane)-poly(2-methyl oxazolyne) (PMOXA-PDMS-PMOXA), and the PS was rose bengal, whose hydrophilicity was increased with bovine serum albumin (BSA). The photodynamic mechanism was investigated by the ERS technique. The photodynamic ROS species generated in the polymersome inner cavity, when irradiated with light, could escape through the permeable-membrane vesicle. The rose bengal- and BSA-loaded polymersomes’ R_g_ and R_h_ values, calculated with SLS and DLS, were 105 ± 2 and 110 ± 2 nm, respectively. Further, the “on-demand” ROS production was evaluated in HeLa and normal cell lines, showing a Trojan-horse-like mechanism as the nanoreactor was toxic only when it was irradiated, generating in situ ROS species during a specific wavelength. Thus, this method was remarked in PDT, as the system combines light and a medication (PS) designed to kill cancer cells or pre-cancerous cells after its activation with light [196].

Chen and collaborators developed cross-linked polyion-complex “PICsomes” based on oppositely charged PEG-based block aniomers and a homocatiomer via vortex mixing of the components. In the assembly process, the amphiphilic Al (III) phthalocyanine chloride disulfonic acid (AlPcS_2a_) PS was encapsulated (AlPcS_2a_-PICsomes), resulting in a drug loading of 11% (*w*/*w*) and 106 nm size by DLS. The fluorescence of AlPcS_2a_ PS was evaluated to analyze its recovery, resulting in an increase of the AlPcS_2a_ fluorescence with the increasing of the laser power with an NIR laser (680 nm). The authors hypothesized that the mentioned fluorescence increase was related to the photoinduced release of AlPcS_2a_ and the subsequent production of ROS within the structure, which disturbed the integrity of the PICsome membrane and increased the permeability. Further, the cellular uptake of AlPcS_2a_-PICsomes was studied with adenocarcinoma human alveolar basal epithelial cells (A549). The results suggested that the clathrin-mediated endocytic pathway was the main pathway of uptake, which confirmed an endo/lysosomal transport for intracellular delivery of AlPcS_2a_. Other results suggested that the AlPcS_2a_-PICsomes were taken up and localized inside the lysosomal compartments and photo-released from the PS and ROS inside cells through the disruption of the lysosomal membrane via the photochemical internalization (PCI) effect. Additionally, the phototoxicity assay reached 50% more cell-growth-inhibitory concentration values than free AlPcS_2a_, which indicated a higher phototoxicity from the AlPcS_2a_-PICsomes [266].

Li et al. proposed light-triggered, clustered polymeric vesicles with self-supplied oxygen for PDT for hypoxic tumor therapy. The vesicles were based on hydrogen peroxide (H_2_O_2_) and poly(amidoamine) (PAMAM) dendrimers conjugated with the PS Ce6/cypate (CC-PAMAM), which were co-assembled with an ABA triblock copolymer containing a ROS-responsive thioketal moiety (hydrophobic segment) and PEG (hydrophilic segment). The assembly of the vesicles (HC@P1-Vesicle) was accomplished through the nanoprecipitation route, obtaining particles with a hydrodynamic diameter of 187.5 nm and a 155.2 nm diameter by TEM with a membrane thickness of around 5.6 nm. The corresponding drug loading levels for the positively charged photoactive agent CC-PAMAM and H_2_O_2_ were 6.1 and 3.2%, respectively. Through the first irradiation at 805 nm, the encapsulated H_2_O_2_ was decomposed into O_2_. In contrast, subsequent irradiation at 660 nm led to disruption of the vesicle through the cleavage of the ROS-responsive moiety, showing a synergistic effect between the self-supply of oxygen and deep tissue penetrability. Moreover, biological studies with human pancreatic BxPC-3 cancer cells showed an extent of ROS-positive cells of 97.3% after 805/660 nm irradiation due to the inside-cell diffusion of H_2_O_2_ and the ^1^O_2_ generated by the CC-PAMAM. The HC@P1-vesicle performed DNA damage and oxidation with a comet tail of 87.26% (alkaline comet) and a lipid peroxidation product MDA level of 6.1 nmol mg^−1^ protein. Moreover, in vivo studies were accomplished in a BxPC-3 pancreatic tumor and confirmed severe hypoxia in the tumor after 24 h post-injection and under repeated 805/660 nm irradiation. Further, the results evidenced a practical PDT effect with poorly permeable tumors due to the penetration capacity of the photoactive CC-PAMAN into hypoxic tumors [267].

Lu and collaborators reported asymmetric polyplex-nanocapsules as assembled polymersomes from the ternary triblock copolymer poly(ethylene glycol)-*b*-polycaprolactone-*b*-poly(ethylene imine) (PEG-PCL-PEI) for the delivery of the hydrophilic endotoxin-free GFP-encoding plasmid DNA (pDNA) for gene transfection and the hydrophobic PS pheophorbide-a (PheoA) for the intracellular PCI effect. The preparation method for the polyplex-nanocapsules was the nanoprecipitation route, with the obtention of structures with a hydrodynamic diameter between 200–280 nm and 200 nm from TEM imaging. The expected organization suggested that the larger PEG block was oriented to the nanocapsules outer layer, and the PEI chain was complexed with the pDNA and located inside the nanocapsules. In contrast, the hydrophobic PCL block conformed to the middle layer in the wall of the nanocapsules, interacting with the PheoA through hydrophobic interactions. The in vitro cellular uptake of YOYO-1-labeled pDNA loaded in the polyplex-nanocapsules was also analyzed. Hence, it was observed that cellular uptake the polyplex-nanocapsules increased the HeLa cells’ fluorescence intensity due to the loaded PheoA. Then, it was proposed that a photochemical-induced endosomal disruption with a 670 nm laser radiation would lead to the escape of the polyplex-nanocapsules from the endo-lysosomes by the ROS produced by the PS, followed by the subsequent pDNA transfection into the cytoplasm. The MTT (3-[4,5-dimethylthiazol-2-yl]-2,5 diphenyl tetrazolium bromide) assays showed a cell viability decrease from 85 to 70% for the PheoA-loaded polyplex-nanocapsules after laser irradiation, which confirmed some phototoxicity in HeLa cells and intracellular gene transfection capacity [268].

The majority of the conventional organic dyes, including the chromophores mentioned above, present notable disadvantages, including photobleaching and the known aggregation caused quenching (ACQ). This concentration-quenching effect, associated with high chromophore or fluorophore concentration, is also related to the intermolecular π–π stacking, especially in those packed with bulky aromatic ring structures, which promote the formation of the aggregates [269,270]. However, diluted solutions involve other issues, such as weak emission, that lead to poor sensitivity and quick photobleaching. Consequently, the aggregation-induced emission (AIE) is a photophysical mechanism where the chromophores in aggregates or in a solid state exhibit luminescence opposite to its molecular constitutes or elementary parts, which lack this property [271,272]. The characteristic performance of AIE luminogens, also called “AIEgens”, has been mainly attributed to the restriction of intramolecular motions (RIM) and restrictions of intermolecular rotations (RIR) and vibrations (RIV). Hence, the nonradiative decay in solution has been associated with the unhindered intramolecular motions, which act as energy acceptors for the excitation energy. The RIM, RIR, and RIV contribute to the disappearance of these energy acceptors, leading to the preservation of the electronic excitation energy responsible for the AIE effect [271,273]. This behavior is evident in supramolecular arrays, such as polymersomes, in which assembly leads to an induced aggregation with the subsequent fluorescence emissions of the AIEgens [81,274].

Moreover, some AIEgens cannot uniquely emit light but generate ROS efficiently, showing dual functions as fluorescence emitter sources and a PSs. Hence, the aggregation-induced generation of ROS (AIG-ROS) may be helpful in PDT, with the robust generation of ROS in the aggregate state [275,276,277,278].

As an example, Cao et al. developed biodegradable fluorescent polymersomes with AIE features and a mitochondria-targeting capacity for enhanced PDT. The biodegradable polymer comprised PEG-*b*-poly(caprolactone-gradient-trimethylene carbonate) (PEG-P(CLgTMC)), with a terminal block of tetraphenylethylene pyridinium-modified PTMC as a functional unit (PAIE) [81]. The tetraphenylethylene pyridinium moiety provided polymersomes with both AIE capacity and positively charged pyridinium-targeting moieties to effectively localize the AIE-polymersomes intracellularly to the mitochondria (Figure 8). Well-defined AIE-polymersomes self-assembled by direct hydration displayed typical AIE behavior. The intramolecular motion was restricted by the non-radiative energy dissipation of the AIEgenic molecules when they were in an assembled/aggregated state. Incorporating hydrophobic BODIPY PS resulted in an intrinsic Förster resonance energy transfer (FRET) between the AIE moieties and BODIPY. AIE-polymersomes showed a targeting capacity toward mitochondria in the A549 lung carcinoma epithelial cell line, and HeLa and HepG2 cancer cell lines compared to a healthy fibroblast 3T3 cell line. Under NIR light irradiation at 660 nm, abundant ROSs were, therefore, generated, leading to fast A549, HeLa, and HepG2 cell necrosis in vitro. In vivo imaging of PS fluorescence demonstrated that the BODIPY loaded in AIE-polymersomes after intratumoral injection was effectively retained in the tumor site using subcutaneous A549 tumor-bearing nude mice as an in vivo model with a strongly boosted therapeutic index.

### 5.4. Controlled Release Induced by the Upconversion Processes

Only a few molecules can respond directly to two-photon NIR irradiation, usually requiring high-intensity and long-duration laser excitation to photoactivate molecules or disrupt the polymersomes and subsequently release the cargo. On the contrary, many photoreactions occur under one-photon UV irradiation, as photons at these wavelengths possess the required energy per photon to break covalent bonds [279] but UV light is phototoxic [235,237]. Strategies based on upconversion (UC) processes have been proposed to overcome the release time limitations associated with NIR and the phototoxic effects of UV irradiation, showing potential in applications related to NIR-light-controlled drug delivery systems. 

The upconversion process is a multiphoton process in which the sequential and not simultaneous absorption of at least two excitation photons leads to light emission at a shorter wavelength than the excitation wavelength. From NP-DDS applications, the low-energy NIR light is converted into high-energy UV or visible light, usually by upconversion nanoparticles (UCNPs).

UCNPs are inorganic nanomaterials based on rare earth elements comprising a crystalline host matrix doped with ions from the fifteen lanthanides. UCNPs can absorb NIR light and convert it into high-energy photons with a wavelength in the UV to visible region. This mechanism is known as the anti-stokes process. Some advantages of UCNP usage are the high penetration depth, no background luminescence interference, multicolor emission, photostability, and they are nonblinking. An outstanding feature of UCNPs is the possibility of upconverting photons from NIR to UV/Visible wavelengths for light-responsive drug delivery processes. Hence, unlike the vast reported light-responsive systems involving UV or short excitation radiation, UCNPs allow the drug release at longer wavelengths. Further, the anti-stokes phenomenon under NIR irradiation leads to shifted visible and UV light emission, whereas the autofluorescence background is minimal, and the light scattering of biological tissues is significantly reduced [280,281,282,283,284,285]. Thus, attaching light-activable moieties onto UCNPs makes upconverting the transducer possible, then absorbs NIR light with the emission of photons with a shorter wavelength and the induction of photoreactions, leading to the triggering of therapeutic agents and/or phototherapy activation. 

Hou and coworkers developed a photodynamic strategy with a nanodumbbell composed of I. hydrophobic NaYF_4_:Yb:Er upconverting (UCN) NPs acting as transducers converting NIR to UV for the production of ROS from the PS zinc (II) phthalocyanine (ZnPc), II. an amphiphilic octadecyl-quaternized poly-glutamic acid (OQPGA) lipid layer as a UCN stabilizer, and III. a polymersome as an external shell based on the amphiphilic PS-PAA and decorated with the arginine–glycine–aspartic acid sequence (RGD peptide) for target delivery (UCN@lipid@PS). The UCN@lipid NPs and UCN@lipid@PS nanodunbbell were prepared through reverse-phase evaporation and nanoprecipitation methods. The obtained UCN presented a 20 nm size, whereas the ZnPc loaded UCN@lipid@PS nanodumbbell presented a diameter of 150 and 195 nm by TEM and DLS, respectively, with a slightly eccentric sphere geometry. During assembly, the encapsulation of ZnPc into the UNC@lipid@PS was accomplished through hydrophobic interactions and evidenced a DLE of 18.03%. The irradiation activated the proposed light-triggered mechanism at 980 nm, which led to the fluorescence emission at 650 nm, activating the PS. In vitro assays with HeLa cells showed lower cytotoxicity (within 500 µg mL^−1^) from the UCN@lipid@PS, as these structures were decorated with the RGD peptide, which binds preferentially to the αvβ3 integrin. Further, the nanodumbbell cell uptake analysis revealed a stronger green upconverting fluorescence and red fluorescence intracellularly, indicating the efficient uptake into HeLa cells. Moreover, the generation of ROS for PDT through a FRET process was hypothesized, considering UCN as donors and PS as acceptors. The HeLa PDT test using the singlet oxygen sensor named 9,10-anthracenediyl-bis(methylene) dimalonic acid (ABDA) evidenced a 62.5% decrease in the fluorescence intensity, confirming the PDT potential of the NPs [286].

### 5.5. Multifunctional Controlled Release

Cancer is a complex disease requiring advanced and effective treatment strategies to overcome the emerging hallmarks of cancer, which is the primary motivation to create multifunctional polymersomes for cancer treatment. Multifunctional polymersomes combine the most notable features from various mono-functional polymersomes. The result is a single polymersome with the best possible combination of favorable functionalities acting in a coordinated manner with one another. Multifunctional polymersomes include (i) the simultaneous incorporation and delivery of two or more therapeutic agents with different mechanisms of action, (ii) the multiple stimulus-response to the release of therapeutic agents, (iii) the combination of two or more therapies, and (iv) the simultaneous incorporation of therapeutic agents and imaging agents. The following subsections consider multifunctional polymersomes belonging mainly to the (ii) and (iii) categories.

#### 5.5.1. Release Controlled by Multiple Stimuli

Polymersomes can use either internal or external stimuli (or both) as the multiple stimulus response to release therapeutic agents. Even a stimulus can trigger reactions to activate other stimuli. Most investigations involve redox and light conditions, as described below. Table 5 reports the design of polymersomes for the release controlled by multiple stimuli.

Liu and coworkers proposed a multi-stimuli-responsive polypeptide-based vesicle assembled through the film rehydration method from the polypeptide poly(S-(o-nitrobenzyl)-l-cysteine) (PNBC) block within amphiphilic copolymer PNBC-*b*-PEO. The vesicles were 83 nm, by DLS, with a membrane thickness of 36 nm. In this study, three secondary structures contributed to the multiple-response performance of the vesicles, (i) the light-responsive ONB moiety, (ii) the oxidable thioether linkers, and (iii) the photocaged redox thiol groups (SH) on the parent poly(l-cysteine) (PLC) backbone. The photocleavage reaction mediated by the hydrophobic ONB group was confirmed in an aqueous solution at 365 nm UV irradiation. After 60 min of irradiation, a morphology transition from vesicles to micelles occurred with the disruption of the π–π interaction in the inner wall of the vesicle and a decrease in the particle’s diameter. Thus, UV light led to the triggered disassembly of vesicles and, further, their reassembly into micelles. This morphological and size transition was also confirmed as a response mechanism related to the thiol groups, contributing to redox sensitivity. Therefore, the polypeptidosomes evidenced a sequential stimuli response to oxidation (H_2_O_2_) and UV irradiation in an aqueous solution. Moreover, DOX was encapsulated, with a drug-loading capacity of 7.7 wt. %. The DOX released from the UV-irradiated particles was 6 times greater for 3 min and reached 91% within 12 h. In comparison, the combination of UV irradiation and H_2_O_2_ oxidation as a synergistic release mechanism presented an on-off release and resulted in a 32% increment of drug release at 12 h and a 14.3% increment at 180 h. Further, the cytotoxicity assays with the HeLa cell line, with the combination of stimuli of UV irradiation for 3 min and H_2_O_2_ oxidation, led to an IC_50_ of 3.80 µg DOX Equiv mL^−1^ lower than the individual oxidation-triggered sample (5.28 µg DOX Equiv mL^−1^) or the non-triggered sample (6.64 µg DOX Equiv mL^−1^) [287].

Zhang and collaborators designed a supramolecular vesicle by host–guest interactions with pH and light responsiveness, consisting of a complex system with cyclodextrins (CD) and AZO molecules [288]. The structure presented a hydrophilic segment composed of β-cyclodextrin with AZO and an acetal (ACE) group and a hydrophobic segment composed of α-cyclodextrin in interaction with the AZO group. Thus, the AZO of the β-CD-AZO-ACE entered the α-CD to form an inclusion β-CD-AZO-ACE⊂α-CD that possessed a hydrophilic cyclodextrin head and a hydrophobically modified tail that self-assembled in an aqueous medium, forming supramolecular vesicles with pH and UV light responses. The AZO isomerization occurred when irradiated with UV light, dividing the inclusion complex and releasing the DOX. Moreover, by decreasing the pH to 5, the acetal groups were hydrolyzed, and two mechanisms could orient the “host–guest” structures, i.e., the change to hydrophilic nature of all the inclusion or the intramolecular inclusion of β-CD-AZO because the cavity of β-CD-AZO has enough space for *trans*-state or even for the *cis*-state of AZO. However, the inclusion of α-CD and β-CD-AZO should be the main formation because α-CD strongly binds to *trans*-state. The LC and EE of the supramolecular vesicles were 1.51 and 15.3%, respectively.

Liu’s group has developed a series of multistimuli-responsive polymersomes with bilayer permeabilization by light-regulated “traceless” crosslinking based on their previous work [248]. In first research, Liu’s group reported photo- and thermo-responsive polymersomes self-assembled from the amphiphilic block copolymer poly(*N*-isopropylacrylamide)-*b*-poly(2-((((2-nitrobenzyl)-oxy)carbonyl)amino)ethyl acrylate) (PNIPAM_31_-*b*-PNBOCA_53_), which was synthesized via consecutive RAFT polymerizations. PNIPAM was a thermoresponsive hydrophilic block with a lower critical solution temperature (LCST) of approximately 32 °C, while PNBOCA with ONB moieties was a light-responsive hydrophobic block. PNIPAM-b-PNBOCA was self-assembled through nanoprecipitation at a lower temperature than the LCST of the PNIPAM blocks (defined as LCST0). The resulting polymeric vesicles collapsed upon temperature rise (T > LCST_0_), and a further temperature increase (T > T_agg,0_) led to the formation of irregular aggregates of collapsed vesicles. The thermo-induced morphological transition from polymeric vesicles to collapsed vesicles was irreversible, whereas the thermo-induced transition between collapsed vesicles and irregular aggregates of collapsed vesicles was likely reversible. The initially hydrophobic PNBOCA bilayers underwent aminolysis-induced cross-linking and a hydrophobic-to-hydrophilic transition upon UV irradiation at 365 nm, resulting in elevated LCST (defined as LCST_uv_). Then, the thermo-induced PNIPAM coronas collapsed (T > LCST_uv_), forming aggregates of cross-linked vesicles (T > T_agg,uv_), and the initial vesicular morphology could be restored when cooling down to lower than LCST_uv_, as opposed to an irreversible morphological transition without UV irradiation (Figure 9a). The co-release of DOX and NR could be regulated by temperature variations and UV irradiation (Figure 9b). Moreover, DO release could be regulated by NIR irradiation at 808 nm in the presence of co-encapsulated PCAs (e.g., ICG). However, the original polymeric vesicles were inherently insensitive to NIR, demonstrating that ICG-loaded polymeric vesicles themselves could be used as a new PTT [289].

Another study reported a photo- and redox-responsive diblock copolymer polymersome synthesized through RAFT polymerization of a coumarin-based disulfide-containing monomer (CSSMA) using a (PEO)-based macroRAFT agent. The amphiphile copolymer comprises reduction-responsive disulfide linkages and photosensitive coumarin moieties in the hydrophobic blocks. The resulting PEO_45_-*b*-PCSSMA_22_ was self-assembled into polymersomes through nanoprecipitation with the encapsulation of DOX and Texas-red-labeled dextran (TR-dextran). Upon irradiation with visible light at 430 nm, the coumarin moieties were cleaved with the generation of highly reactive primary amine groups, which spontaneously underwent protonation, intramolecular acyl migration, and inter/intrachain amidation reactions with the ester moieties, thereby cross-linking and permeating the bilayer membranes. This process only released DOX, while TR-dextran was retained within the cross-linked polymersomes that preserved their integrity. The TR-dextran release was only achieved in the presence of glutathione (GSH), which cleaved the disulfide linkages, disintegrating the polymersomes. The dual-stimuli-responsive polymersomes enable sequential release of small and large molecules [290].

Tong and coworkers designed dual photo- and pH-responsive polymersomes self-assembled from a host–guest complex between a water-soluble pillar[6]arene (WP6) and an AZO-ended functionalized PCL (PLC-Azo) [291]. Dual responsive polymersomes were self-assembled by nanoprecipitation. The AZO moiety and WP6 possess photo- and pH-responsiveness, respectively. The photoisomerization of the AZO group could control the host–guest complex. When polymeric vesicles were irradiated at 365 nm, they changed into solid nanospheres since *cis*-AZO isomer could slide out from the pillar cavity [6]arene because of the size mismatch. The disassociated system could be reformed by exposing it to visible light at 435 nm. Moreover, the vesicles translate to irregular aggregates by lowering the pH (pH = 2.2) and reformed at pH = 7.4. The reversible transformations between vesicles and solid aggregates (spheres or irregular aggregates) under UV or pH stimulus were utilized for controlled DOX release. UV stimulus showed a lower cumulative release than the pH stimulus. The polymeric vesicles presented excellent cytocompatibility toward HepG2 cells and can be further applied for the controlled release of DOX.

Weng and coworkers developed a novel amphiphilic triblock photo-responsive self-reducible polymer (PRSRP) that was self-assembled in an aqueous solution [292]. The polymer was built by incorporating a photo-locked “attacker” (ONB protected dithiothreitol, ONB-DTT) and an L-cystine derived “acceptor” into the same polymer “unit.” The PRSRP could respond under reductive degradation, such as intracellular glutathione (GHS) levels, and the disulfide linkages (“acceptors”) within the backbone could still be cleaved by the ONB (“attackers”) unlocked by UV-light. Further, the DTT-attackers were generated in situ and could overcome the steric hindrance, facilitating the reducing agents’ penetration (Figure 10). The amphiphilic triblock resulted from the co-polymerization of ONB-DTT with L-cystine dimethyl ester diisocyanate (CDI) and methoxyl PEG (mPEG). The presence of self-assembled vesicle-like structures in an aqueous solution was evidenced by their hydrodynamic and gyration ratios, with a related value, ρ, of 1.02, confirming vesicles obtention and an average diameter of 90 nm. The authors demonstrated that the PEG block was located in the corona and interior, whereas the CDI and ONB-DTT groups aggregated within the membrane. Moreover, one hydrophilic (DOX) and one hydrophobic (FITC) molecule were encapsulated and tested in MCF-7 breast cancer cells, evidencing an effective antitumor therapy. Wang et al. developed both UV light and reductive milieu polymersomes to conduct in situ transitions to polyion complex vesicles (PICsomes) accompanied by switching vesicle bilayer permeability. Thus, polymersomes were self-assembled from amphiphilic block copolymers containing monomers with carbamate linkages (DPA and DEA) and caged carboxyl comonomers ONB ester photo-caged carboxyl monomer (NCMA) or disulfide-caged carboxyl monomer (DCMA) located in the stimuli-responsive initially hydrophobic block. Therefore, two series of carbamate-containing monomers (four types of monomers and corresponding block copolymers) with caged carboxyl and tertiary amine were synthesized. Whereas monomers with NCMA and DCMA generated carboxyl moieties upon actuation of UV irradiation and reductive milieu, respectively, the second series contained pH-responsive DPA and DEA monomers with both a carbamate linkage and a tertiary amine. The authors proposed that electrostatic interactions between the amines and carboxylic acid created electrostatic interactions enhanced by side-chain hydrogen bonding. Before UV irradiation occurs, polymersomes are stabilized by hydrophobic interactions. Then, ion-pair interactions were generated within vesicle bilayers related to the light-actuated polymersome-to-PICsome transition with UV light. Moreover, a reductive milieu could also trigger the polymersome-to-PICsome transition, considering applications in the complex biological milieu. Further, gemcitabine, DOX, 5-fluorouracil (5-FU), and calcein were encapsulated as a proof of concept.

Tsai et al. developed photo- and redox-responsive polymersomes for cancer chemotherapy [293], where polymersomes were assembled by the double emulsion method from the synthetized amphiphilic diblock copolymer poly(ε-caprolactone)-ONB-SS-poly(methacrylic acid) (PCL-ONB-SS-PMAA). The two polymeric chains were linked through a responsive ONB ester next to a GSH-responsive disulfide linkage (SS). Hydrophobic core−shell UCNPs (NaYF4:Yb/Tm (core)/NaYF4 (shell)) and DOX were simultaneously encapsulated into the polymersomes during assembly. DOX was encapsulated into the hydrophilic core, and the hydrophobic core−shell UCNPs were loaded into the hydrophobic bilayer. UCNPs emit in situ UV light around 365 nm under NIR light irradiation at 980 nm, inducing the photorupture of the ONB linkage. By combining ONB linkage-induced photorupture and GSH disulfide cleavage, it enhanced DOX release for chemotherapy. Polymersomes containing core−shell UCNPs and DOX (UCNP-PNSP@DOX NPs) were cytotoxic against three lung cancer cell lines (A549, CR-5802, and HEL-299 cells) under the assistance of a 980 nm diode laser. UCNP-PNSP@DOX NPs also inhibited tumor growth in A549 tumor-bearing mice under 980 nm diode laser irradiation compared with those without laser irradiation and those treated with free DOX.

#### 5.5.2. Controlled Release for Multiple Therapy Polymersomes

Combining two or more therapeutic agent release mechanisms induced by light leads to integrating two or more therapy types (e.g., chemotherapy, PTT, and PDT) for light-responsive polymersomes. This multifunctional controlled release mechanism is exciting, mainly when resistance to therapeutic processes exists. Table 6 summarizes the polymersomes designed for the controlled release of multiple therapies.

Liao and coworkers fabricated light-responsive polymersomes for the co-release of AuNRs and DOX in a combined PTT-chemotherapy. AuNRs and DOX were co-encapsulated in the hydrophilic core of polymersomes self-assembled from mPEG-PCL using a double emulsion method. The release of DOX could be readily controlled with NIR irradiation at 808 nm and pH control. The heat produced by AuNRs under NIR irradiation was absorbed partly by polymersomes, which induced their disruption because the temperature was higher than the melting point of mPEG-PCL. Moreover, heat from AuNRs increased cytotoxicity against C26 tumor cells due to the photothermal effect. Effective C26 mouse tumor ablation was observed after intravenous injection of polymersomes followed by NIR irradiation [294].

He et al. reported photoconversion-tunable fluorophore polymersomes for wavelength-dependent NIR-induced cancer therapy that facilitated PDT under 660 nm irradiation or PTT under 785 nm irradiation. This dual therapy is possible by introducing the BODIPY fluorophore (Figure 11). BODIPY was encapsulated in the hydrophobic membrane of polymersomes self-assembled from terpolymer PEG_45_-PCL_60_-PNIPAM_33_ under ultrasonication. After being assembled within polymersomes at a high drug LC (20%), BODIPY molecules aggregated in both the J-type and H-type conformations, causing the red-shifted absorption into the NIR region, low radiative transition, and excellent resistance to photobleaching. Under irradiation at 660 nm, the polymersomes mainly displayed excellent singlet oxygen quantum yield through the ^3^O_2_-to-^1^O_2_ transition, leading to abundant intracellular ^1^O_2_ at the tumor for effective PDT. When polymersomes were irradiated at 785 nm, they mainly possessed photothermal conversion efficiency through non-radiative transition and mild ^1^O_2_ generation, thus leading to potent hyperthermia at the tumor for robust PDT-synergized PTT. Moreover, polymersomes showed remarkable cellular uptake in 4T1 murine tumor cells via clathrin-mediated endocytosis-enhancing tumor accumulation. Effective cytoplasmic drug translocation occurred through ^1^O_2_-mediated lysosomal disruption. In vivo studies in mice bearing 4T1 tumors under irradiation at 785 nm showed that polymersomes have an excellent ability to generate potent hyperthermia for effective PTT treatment due to their preferable accumulation at the tumor and enhanced photothermal conversion efficiency. In comparison, the 660 nm irradiation only caused poor hyperthermia. Thus, with no regrowth, polymersomes effectively achieved tumor ablation through PDT-synergized PTT under 785 nm irradiation [295].

Zhu and collaborators developed nano-sized bubble-generating polymersomes for triggered drug release and synergistic chemo-photothermal combined therapy (BG-DIPS). The block copolymer consisted of PCL_8000_-PEG_8000_-PCL_8000_ and the assembly was by the thin-film rehydration method. Thus, ICG, a NIR clinical imaging agent, was encapsulated in the hydrophobic membrane of the vesicles. DOX was encapsulated into the hydrophilic inner cavity of the polymersomes using the ammonium bicarbonate (NH_4_HCO_3_) gradient loading method. In addition, NH_4_HCO_3_ loaded into the polymersomes can trigger quick drug release in response to hyperthermia or acidic pH values. The BG-DIPS were stable in blood circulation, relying on the outer PEG layer, and could accumulate at the tumor site via the EPR effect. Moreover, the stability in the physiological conditions, estimated through ζ, was -16.4 ± 0.5 mV, favorable for reducing plasma protein adsorption and opsonization achieving long circulation in blood. These structures presented an efficient inhibition in 4T1-Luc tumor growth due to hyperthermia and chemotherapy effects, with EE and LC of 14.55 ± 1.61 and 3.49 ± 0.21%, respectively. The EE and LC of ICG in BG-DIPS were calculated to be 82.32 ± 1.37 and 4.23 ± 0.18%, respectively [296].

Liu’s group had also designed polymersomes for multiple therapies. They reported dual chemo-thermal therapy plasmonic polymersomes using functionalized amphiphilic hybrid AuNPS with the diblock copolymers poly-(ethylene oxide)-*b*-poly(2-(methacryloyloxy) ethyl 5-(1,2-dithiolan-3-yl)pentanoate) (PEO-*b*-PMALA) and poly(2-((((2-nitrobenzyl)oxy)carbonyl)amino)ethyl methacrylate)-*b*-poly-(2-(methacryloyloxy)ethyl 5-(1,2-dithiolan-3-yl)pentanoate)) (PNBOC-*b*-PMALA), both synthesized via RAFT polymerization. The grafted polymer chains enabled the self-assembly into polymersomes. The PMALA segment was attached to the surface of the AuNPs, forming a multivalent Au-S bond. In contrast, the PEO (hydrophilic) and PNBOC (hydrophobic and photoreactive) contributed to the amphiphilicity of the conjugated particles localized in the corona and within the membrane, respectively. Two structures varying the AuNPS size (3 and 13 nm) and the degree of polymerization of the PEO and PNBOC blocks (PV-1 and PV-2) were used for the assembly where also the hydrophilic DOX and the hydrophobic paclitaxel (PTX) active components were encapsulated through nanoprecipitation. The mechanism of drug release was promoted via irradiation with visible light (410 nm) for the cross-linking of the membrane through primary groups generated by the NBOC moieties due to the amine inter/intra amidation reactions, leading to a hydrophobic-to-hydrophilic transition with the subsequent release of the active components. The plasmonic vesicles were also proposed as potential computer tomography contrast agents for theragnosis. The structures obtained by the two types of nanoprecipitations evidenced hydrodynamic diameters of 160–320 and 194 nm for PV-1 and PV-2, respectively. The PV-2 vesicles showed a release of 83.7 (DOX) and 85.3% (PTX). Further, in vitro studies with PV-2 interacting with HepG2 cells evidenced a chemotherapeutic effect with a decrease in the cell viability to 5% after irradiation. Moreover, the photothermal activity conferred by the AuNPs with a temperature rise of 9.5 °C under 560 nm irradiation synergistically contributed to the therapeutic effect [297].

Tang et al. developed light-responsive polymersomes through self-assembling poly(propylene sulfide)-PEG (PPS-PEG) encapsulating a heat-sensitive DOX prodrug (CED_2_) by the conjugation of two molecules of DOX onto the photothermal croconaine dye (CR780), using the Edman linker. The membranes and the inner spaces of the vesicles were loaded with the CED_2_ prodrug and the hydrophilic free radical precursor 2,2′-azobis[2-(2-imidazoline-2-yl)propane] dihydrochloride (AIPH), respectively. The nanovesicles co-encapsulating both active components were called (v-A-CED_2_). The activity of four interrelated processes controlled the release mechanism. They were, (i) hyperthermia generated by the CR780 under NIR 808 nm irradiation as the external input, (ii) acidic media from the tumor microenvironment which transformed the prodrug into DOX (chemotherapy), (iii) production of ROS by AIPH decomposition through the heat generated with the irradiation, and (iv) the oxidation of the ROS-responsive amphiphilic copolymer PPS-PEG, leading to the degradation of the vesicle for the release of DOX. The nanovesicles presented a 100 nm hydrodynamic diameter with a membrane thickness measured by TEM between 6 and 8 nm. Further, the DLC and LE for the CED_2_ were 22.4 and 83.7%, respectively, whereas the same parameters for AIPH were 3.7 and 46.2%, respectively. In vitro studies showed 74.8% cell death, whereas in vivo studies demonstrated suppression of tumor growth in a xenograft model with a rate of 97% [298].

Zhang and collaborators proposed a nanosized hybrid polymersome based on a combination of PEG-*b*-PLA diblock copolymers and the phospholipid 1,2-dioleoyl-sn-glycero-3-phosphocholine (DOPC) for enhancing biocompatibility. The hybrid vesicles encapsulated porous silicon NPs (Psi) (150 nm) conjugated with AuNRs (50 nm length) as composite nanoparticles (cNPs) for photothermal functionality. Further, the cNPs could be loaded with different drugs via adsorption. The assembly of the polymersomes encapsulating cNPs in the core-shell was carried out through double emulsion (w/o/w) employing microfluidic techniques. Moreover, three hydrophobic anticancer drugs, docetaxel, rapamycin, and afatinib, were simultaneously encapsulated with an encapsulation efficiency of 92% and a drug loading capacity of 20% for encapsulated cNPs with single or combined drugs. The hydrodynamic diameter measured by DLS was 286 nm, and DOX was also encapsulated. Over 90% of DOX and 15% rapamycin were released within 30 min under NIR light (808 nm) compared to 20 and 5% without laser, respectively. Additionally, in vitro assays with different cell lines confirmed the synergistic effect of drug combinations that enhanced cell death and cytotoxicity. Moreover, the photothermal effect combined with two drugs (decetaxel and afatinib) enhanced the effectiveness against SKBR-3/AR cells, reaching up to 80% cell death in 30 min of incubation. Further, an in vivo study with a HER2-positive breast cancer mouse model showed that the triple combination of afatinib, docetaxel, and rapamycin, suppressed 94.6 and 87.5% of the tumors with dosages of 5 and 2.5 mg kg^−1^, respectively. Thus, the study demonstrated the potential of the polymersomes as multifunctional nanovehicles for chemotherapy and PTT [299].

Saravanakumar et al. designed polymersomes for NIR-light-controlled combined chemo-PDT and -PTT [300], employing PEG-block-poly(β-aminoacrylate)-block-PEG) (mPEG-*b*-PBAC-*b*-mPEG) ABA-type amphiphilic triblock copolymers. The ABA copolymer was assembled through the nanoprecipitation method with the co-encapsulation of the hydrophobic NIR PS IR-780 into the hydrophobic membrane and DOX into the hydrophilic core. IR-780 dye used NIR light to convert the NIR photon energy into heat while generating ^1^O_2_ from ^3^O_2_, simultaneously enabling photodynamic and photothermal activity. Moreover, ^1^O_2_ mediated the disassembly of the polymersomes via ^1^O_2_-photocleavage of PBAC, releasing the chemotherapeutic DOX. Thus, the designed polymersomes combined the synergistic effect of photothermal, photodynamic, and chemotherapy. The biological evaluation of the polymersomes co-encapsulating IR780-DOX under NIR light irradiation exhibited remarkable cytotoxicity in the 4T1 and CT26 cell lines compared to empty polymersomes.

He’s group proposed oxidative-responsive polymersomes mediated by NIR irradiation that exhibited the PDT-chemotherapy combination effect [301]. For the self-assembly of polymersomes, block copolymer PPS_20_-*b*-PEG_12_ was self-assembled through solvent dispersion. Hydrophobic zinc phthalocyanine (ZnPc) and DOX were co-encapsulated as PS and chemotherapeutic agents. Under NIR irradiation at 660 nm, ZnPc generated ^1^O_2_, which oxidized the neighboring sulfur atoms on the PPS block. The oxidation induced the breakdown of polymersomes, leading to the release of encapsulated DOX. The released DOX and the production of ^1^O_2_ showed an excellent anti-tumor effect by synergistic PDT-chemotherapy. In vivo studies showed that combined PDT-chemotherapy could efficiently accumulate in nude mice bearing malignant melanoma (A375 cells), thus leading to significant inhibition of tumor growth compared to individual therapies.

Hu et al. developed light-responsive polymersomes assembled from PEG-PCL as a strategy to maximize the synergistic efficacy of PDT and PTT for prostate cancer therapy [302]. AuNRs stabilized with cetyl trimethyl ammonium bromide (CTAB) were employed to coordinate ICG in the hydrophobic membrane of polymersomes. AuNRs were used as PCAs, while ICG was used as a bifunctional PS and PCA. PDT and PTT occurred under the same NIR wavelength (785 nm). The generation of ROS induced PDT and destroyed the integrity of the lysosomal membrane, promoting the translocation of ICG and AuNRs into the cytosol. Moreover, the double PTT produced by both ICG and AuNRs could engender more significant damage to the tumor cells because of the close distance to organelles. These polymersomes promoted the apoptosis of PC3 tumor cells, originated from a prostate-specific membrane antigen (PSMA)-negative castration-resistant subtype. Furthermore, the in vivo experiment confirmed the enhanced PTT/PDT efficacy of AuNR/ICG polymersomes under NIR irradiation.

Sajadi et al. designed a photoresponsive amphiphilic copolymer of acrylate porphyrin and acrylate β-cyclodextrin (β-CD) from AZO-containing dextran as the initiator through nanoprecipitation in aqueous media. The amphiphilic block copolymer b-(AZO-grafted Dex)-b((MMA)-r-(β-CDAc)-r-(PorAc)) self-assembled into polymersomes with an approximately 200 nm size. Hence, the host/guest mechanism was explored from the supramolecular interaction between AZO and the β-CD rings, which were also involved in the reversible photoresponsive mechanism and the entrapping and releasing of molecules by UV light irradiation. The AZO group, as the host molecule, controlled the structure and drug release from the polymersomes. In contrast, porphyrin PS molecules within the polymeric bilayer formed H and J aggregates, increasing the stability under long-term UV irradiation, acting as noncovalent cross-linking points, and enhancing the photodynamic behavior. The ability of porphyrins containing polymersomes to generate singlet oxygen was evaluated with ICG as a reactive species indicator. The single/dual encapsulation efficiency was evaluated with 5-FU and quercetin (Q), showing a single-drug efficiency of 53.1 and 35.5%, respectively. The dual-drug encapsulation efficiency was 32.6 for Q and 37.3% for 5-FU. The results also showed that the 5-FU release was higher than Q due to its hydrophilic character [200].

In an example of simultaneous photoconversion with UCNPs and photocleavage for multiple therapies, He and collaborators developed a nanoreactor based on oligo(ethylene glycol) monomethyl ether methacrylate (OEGMA) mixed with synthetically derived eosin Y (derived EoS) and ONB oxycarbonyl aminoethyl methacrylate (NBOC) hydrophobic monomer with an ONB photolabile group (Figure 12) [303]. The resulting amphiphilic diblock copolymer P(OEGMA-co-EoS)-b-PNBOC (POPN) was synthesized via RAFT polymerization. Then, polymersomes were obtained by the solvent exchange method. The polymersomes’ membrane permeability for releasing the hydrophilic prodrug banoxantrone dihydrochloride (AQ4N) for topoisomerases inhibition was modulated by NIR light irradiation. It interacted with Tm^3+^ (2%), Er^3+^ (0.2%), Gd^3+^ (10%), and Yb^3+^ (20%) co-doped UCNPs emitting a higher energy light in the UV region. The interaction promoted a destabilization reaction (tuning permeability and maintaining its integrity) and activated the EoS PS from the structure’s core. The EoS promoted a photodynamic effect related to ROS production that increased the hypoxic condition inside cancer cells, which subsequently promoted the enhanced activity of the hypoxia-activating prodrug AQ4N during 24 h. The EE and LC were calculated to be 98 and 49% (*w*/*w*), respectively. The polymersomes were also functionalized with glycyrrhetinic acid (GA), which targets protein kinase C (PKC) α, which is overexpressed on the tumor cell membrane, and interacted with the mitochondrial respiratory chain. The GA-coated polymersome showed an impressive cell proliferation suppression of 73 ± 2.8% by NIR irradiation in a hypoxic environment.

Some authors have integrated the strategies of multiple stimuli and therapies. For example, Wang et al. combined PTT-chemotherapy using reduction and pH dual-sensitive polymeric vesicles encapsulating DOX and decorating with a dense, continuous gold nanoshell. First, a reduction and pH dual-sensitive polymeric vesicle was self-assembled from the diblock copolymer polyethyleneimine-*b*-poly(2-diisopropylamino/2-mercaptoethylamine) ethyl aspartate PEI-PAsp(DIP/MEA) by the emulsion method with the simultaneous encapsulation of DOX. Then, a leak-tight gold nanoshell with good NIR light-to-heat conversion was grown on the polymeric vesicle surface via the in situ Au(Ⅲ) reduction approach. Upon NIR irradiation at 808 nm, the gold nanoshell was ruptured, increasing temperature and promoting PTT. Additionally, in the presence of GSH at pH 5.0, the disulfide bond was broken, resulting in polymeric vesicle disassembly and DOX release. The efficiency of combined PTT-chemotherapy was evaluated in vitro with the human hepatoma Bel-7402 cell line and in vivo in nude mice bearing human Bel-7402 hepatoma after intravenous administration [304].

Liu’s group designed polyprodrug vesicles for synergistic PDT-chemotherapy. Amphiphilic polyprodrugs of poly(N,N-dimethylacrylamide-co-Eosin Y)-b-polycamptothecin P(DMA-co-EoS)-b-PCPTM were synthesized via RAFT polymerization. An oil-in-water (O/W) emulsion method was used to self-assemble the amphiphilic polyprodrugs into polymeric vesicles. Hydrophobic oleic acid (OA)-stabilized NaYF4:Yb/Er UCNPs were introduced during the self-assembly process. Upon NIR irradiation a 980 nm, UCNPs transferred energy to EoS PS with the generation of ^1^O_2._ In situ generated ^1^O_2_ could exert its PDT effect, disrupt the membranes of endolysosomes, and, thus, facilitate the endosomal escape of uptake polymersomes by PCI. The abundant GSH within the cytosol then triggered the release of CPT via the cleavage of disulfide linkage, thereby activating the chemotherapy process. The synergistic PDT-chemotherapy was studied in the HepG2 cell line [305].

Lei et al. prepared polymeric vesicles to combine PDT-chemotherapy as a promising strategy to achieve an enhanced anticancer effect. Polymeric vesicles were self-assembled from poly(L-lactic acid)-*b*-poly(sodium 4-styrenesulfonate) (PLLA-*b*-PSS, BCP for short) by nanoprecipitation with the encapsulation of DOX. After polymeric vesicle formation, ferric citrate (Cit-Fe(III)) was attached to the surface of polymeric vesicles through the electrostatic interaction and zeolitic imidazolate framework 8 (ZIF-8) growth among the surface of the particles, which yielded the nanocomposite BCP/Cit-Fe(III)@ZIF-8. Cit-Fe(III) catalyzed ROS generation, such as ^•^OH and sulfate radicals (SO_4_^•−^), upon H_2_O_2_ and visible light stimuli. The generation of SO_4_^•−^ (due to the oxidation of −SO_3_^−^ in the PSS block) disassembled polymeric vesicles, enabling the gradual diffusion of DOX into the ZIF-8 channels. A burst of DOX release was achieved by the collapse of ZIF-8 under low pH conditions. As shown, the nanocomposite only combined chemotherapy and phototherapy in the presence of H_2_O_2_ and pH stimuli upon visible light exposure. The prepared DOX-loaded nanocomposite exhibited good selectivity for generating ROS and releasing the drug in MCF-7 cells instead of 3T3-Swiss Albino cells (embryo fibroblast and normal cell) due to the higher concentration of H_2_O_2_ and lower pH in tumor cells [306].

Recent advances regarding the implementation of polymersomes correspond to the development of structures with motility, such as micro-nanomotors [307,308]. In our group, Mena-Giraldo and Orozco have recently developed a Janus UV-photosensitive polymeric chitosan micromotor (J-PCM), demonstrating the enzymatic protection against UV light using UV-absorbing molecules. Thus, the polymer was synthesized by covalently linking azobenzene molecules to chitosan, and the J-PCM assembly was accomplished via reverse micelles. Simultaneously, the micro-motors co-encapsulated magnetite and Pt NPs as the mechanism for the anisotropic formation step. Remarkably, photo-isomerizable azobenzene molecules demonstrated absorption and UV-light protection with the immobilized Lac and Cat enzymes, among other proteins. Further, magnetic and catalytic Pt NPs contributed to motion and dynamically improved the enzymatic activity and substrate degradation processes [308].

Hence, unlike Brownian motion, micro and nanomotors have enhanced motility by interacting and converting energy from milieu stimuli into controlled movement [309]. Thus, authors such as Cao et al. developed light-responsive phototactic/phototherapeutic nanomotors (polymersomes) that are able to move from the interaction with NIR light. The polymersomes were based on a biodegradable (PEG)-PTMC(TPEDC) amphiphilic copolymer modified with a second-generation AIE moiety comprising tetraphenylethylene and dicyanovinil moieties (TPEDC). The motor size was from 300 to 500 nm, assembled via nanoprecipitation. The AIE moieties emitted fluorescence and enabled ROS production as a PDT. Moreover, the subsequent coating of the polymersomes with an Au nanoshell layer allowed the activation through two-photon NIR irradiation and the translating of radiative energy into the physical driving force leading to self-propulsion. Therefore, both AIE and Au materials contributed to the photothermic fluorescence and the LSPR as an energy supply and sink, respectively, to promote a thermophoresis mechanism (Figure 13). Furthermore, the nanomotor’s phototaxis via thermophoresis was evaluated as a mechanism for membrane percolation as a cellular interaction pathway and uptake. Thus, the interaction between nanomotors and the biological model with HeLa cancer cells presented a localized percolation [278]. Therefore, this novel investigation proposed the polymersomes as nanomotors boosted by light with a non-chemical fuel, simultaneously including a new topic as AIEgenic nanomaterials with motile properties and PDT potential.

Examples of theragnostic applications also are reported in the literature, in which the capabilities for therapy and diagnosis are integrated into one only polymersome. Following its work, Liu’s group designed NAD(P)H:quinone oxidoreductase isozyme 1 (NQO1)-responsive polymeric vesicles covalently conjugated with the PSs coumarin and Nile blue for integrated theragnostic functions. Polymeric vesicles were self-assembled from amphiphilic block copolymers containing quinone trimethyl lock-capped self-immolative side linkages and quinone-bridged PSs in the hydrophobic block. After the aqueous self-assembly of amphiphilic block copolymers via the nanoprecipitation approach, the polymeric vesicle surface was functionalized with cRGD-targeting peptides and a fluorescent dye. Initially, fluorescence emission and PDT potency were in an “off” state due to “double quenching” effects, i.e., dye-aggregation-caused quenching and quinone-rendered photoinduced electron transfer (PET) quenching. Upon cellular uptake, intracellular NQO1 triggered self-immolative cleavage of the quinone trimethyl locks, PS release, and simultaneous NIR emission and PDT activation (Figure 14). This process was accompanied by the transition of nanostructure morphologies from polymeric vesicles into cross-linked micelles with hydrophilic cores and smaller sizes and triggered dual drug release, which was directly monitored by enhanced magnetic resonance (MR) imaging for polymeric vesicles conjugated with a DOTA(Gd) complex in the hydrophobic bilayer. Cellular uptake and triggered cargo release were studied in NQO1-positive A549 cells and HeLa cells. In vivo tumor inhibition of NQO1-responsive polymeric vesicles was examined in A549 tumor-bearing mice. The theragnostic potency of NQO1-responsive polymeric vesicles was also evaluated with in vivo fluorogenic imaging of tumor-bearing mice [310].

## 6. Concluding Remarks and Outlooks

According to the examples reviewed in the previous section, light-responsive polymersomes are ideal candidates for smart nanocarrier-based DDSs, also known as SDDSs. The “smart” characteristic has been proposed as a conceptualization of “strategies and materials to advance and refine treatments” [52]. Nowadays, SDDSs consist of NPs able to carry anticancer therapeutic agents to the tumor site, targeting mechanisms to locate the tumor site, and stimuli-responsive nanocarriers to release the anticancer therapeutic agents at the pre-located tumor cells [31,42,311]. It could be noted that light-responsive polymersomes possess unique features that can be modified through self-assembling amphiphilic copolymers with various chemical functionalities to obtain SDDSs. Depending on the chemical composition of the amphiphilic copolymer and their functionality, polymersomes can be turned into intelligent designs to obtain NP-DDSs with low aggregation rates, high stability against the biological environment, and long circulation half-lives, among other features. The functional groups also enable the easy surface modification with ligands for guiding therapeutic agents to the tumor site via passive or active targeting. With these features, polymersomes reach the tumor site before their clearance and degradation, fulfilling the first and second requirements of SDDSs. For the third requirement, light induces the disassembly and disruption of polymersomes, releasing therapeutic agents in a spatiotemporally controlled manner.

In addition to these smart features, light-responsive polymersomes can carry a variety of hydrophobic and hydrophilic cargoes for cancer treatment in separated compartments, ranging from conventional therapeutic agents, proteins, enzymes, or DNA, to smaller nanoparticles, such as metal or UCNPs. Hence, the co-encapsulation of multiple anticancer therapeutic agents or multi-stimuli response moieties and high cargo-retention efficiency may lead to effective multifunctional cancer treatments with low doses. Furthermore, light-responsive polymersomes can also integrate two or more therapy types (e.g., chemotherapy, photothermal, and photodynamic therapies) to avoid or minimize drug resistance. Moreover, polymersomes can be designed so that the therapeutic effect relies only on cancer cells and tissues. Moreover, polymersomes can simultaneously encapsulate therapeutic and diagnostic agents for theragnostic applications towards personalized medicine.

Despite the advantages of the mentioned features, there are still particular challenges for polymersomes as SDDSs, e.g., the toxicity of nanocarriers is one of the main limitations to face. In this context, polymersomes must be stable and release appropriate amounts of anticancer therapeutic agent to the tumor cells for a sufficient time, after which they must be degraded entirely and cleared out from the body. In this regard, polymersome toxicity on vital organs should be better assessed. Besides, the successful translation to the clinic requires a large-scale assembly with good reproducibility. Therefore, significant efforts must be made towards designing and synthesizing amphiphilic copolymers and the subsequent scaling up of industrial assembly procedures.

On the other hand, biological considerations and the understanding of nanocarrier–biological environment interactions are not obvious. Moreover, some challenges of DDSs, such as the abrupt release (“burst release”) post-administration of a large fraction of adsorbed drug, the complexity to encapsulate drugs that are poorly miscible, and also the poor drug LC, need to be taken into account [174]. More studies on the therapeutic advantages of polymersomes in the biological milieu and in vivo investigations comparing bioengineered liposomes and FDA-approved formulations are needed. This practice aims to establish an actual equivalence between the theoretical properties of polymersomes and their real performance. Yet, to our knowledge, there are no comparative studies about the advantages of one nanocarrier over the others.

## Figures and Tables

**Figure 1 nanomaterials-12-00836-f001:**
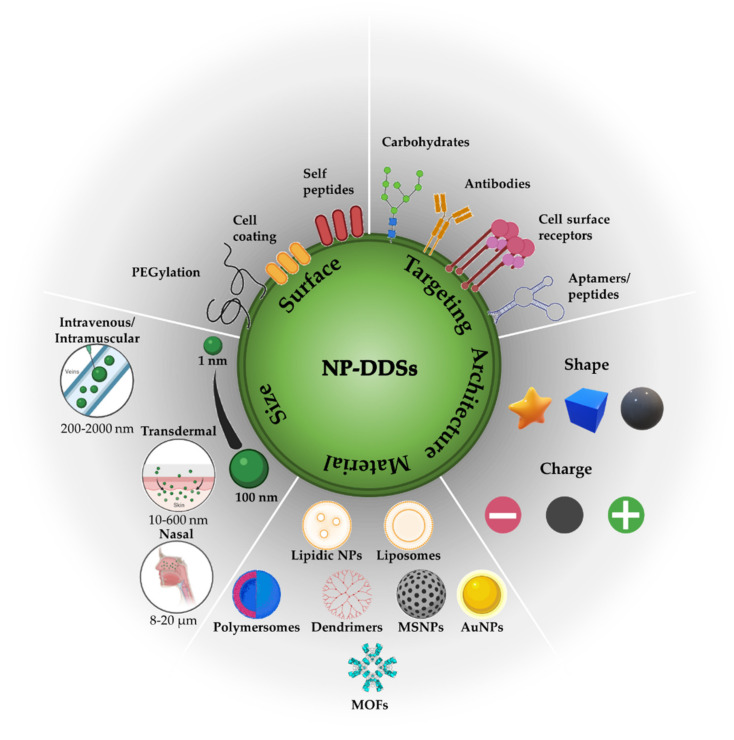
Biochemical and physicochemical properties of NPs for enhanced DDSs.

**Figure 2 nanomaterials-12-00836-f002:**
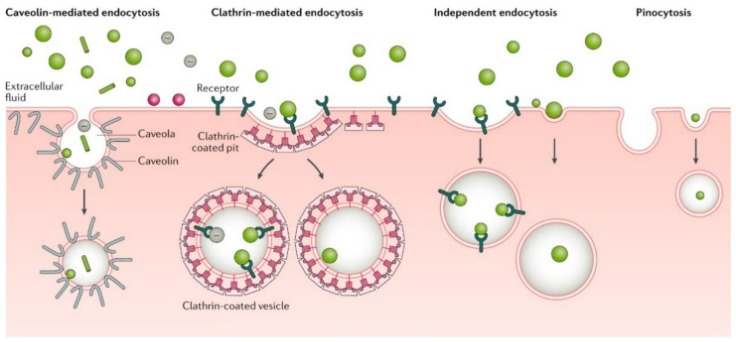
Pathways for cellular uptake mediated by endocytosis. Reprinted with permission from [45]. Copyright © 2022, Springer Nature Limited.

**Figure 3 nanomaterials-12-00836-f003:**
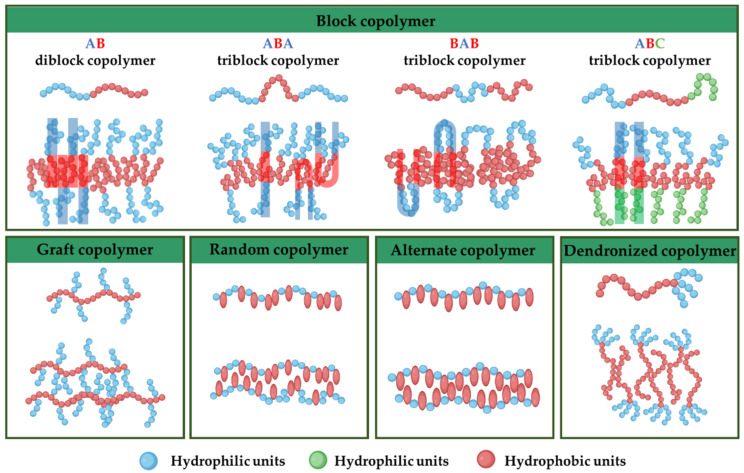
Schematic representation of different amphiphilic copolymers and their proposed spatial organization in the bilayer membrane of polymersomes.

**Figure 4 nanomaterials-12-00836-f004:**
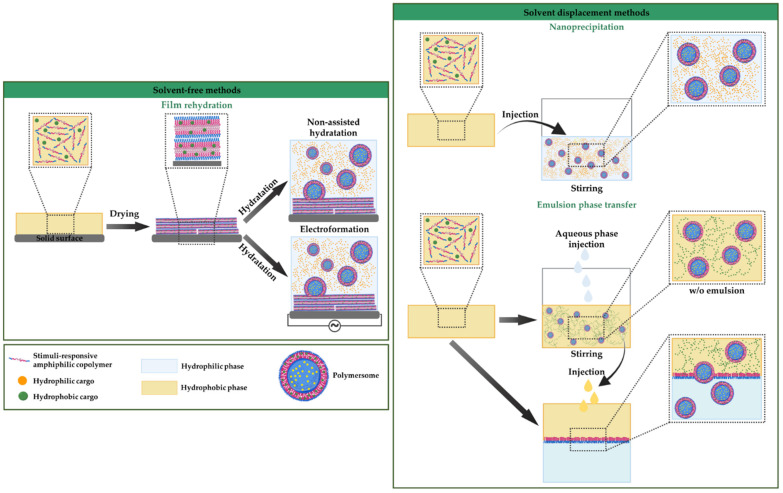
Schematic representation of self-assembly methodologies of polymersomes and encapsulation via the during-self-assembly processes, based on [70].

**Figure 5 nanomaterials-12-00836-f005:**
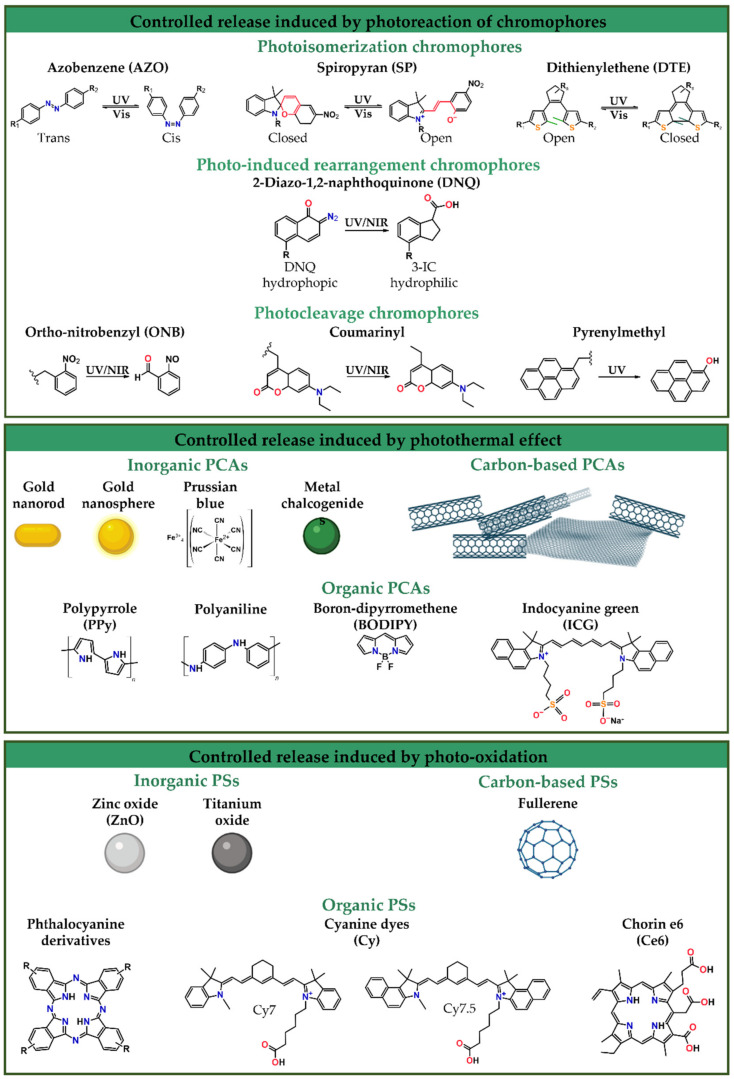
Light-activable moieties according to the release mechanism of the light-responsive polymersomes.

**Figure 6 nanomaterials-12-00836-f006:**
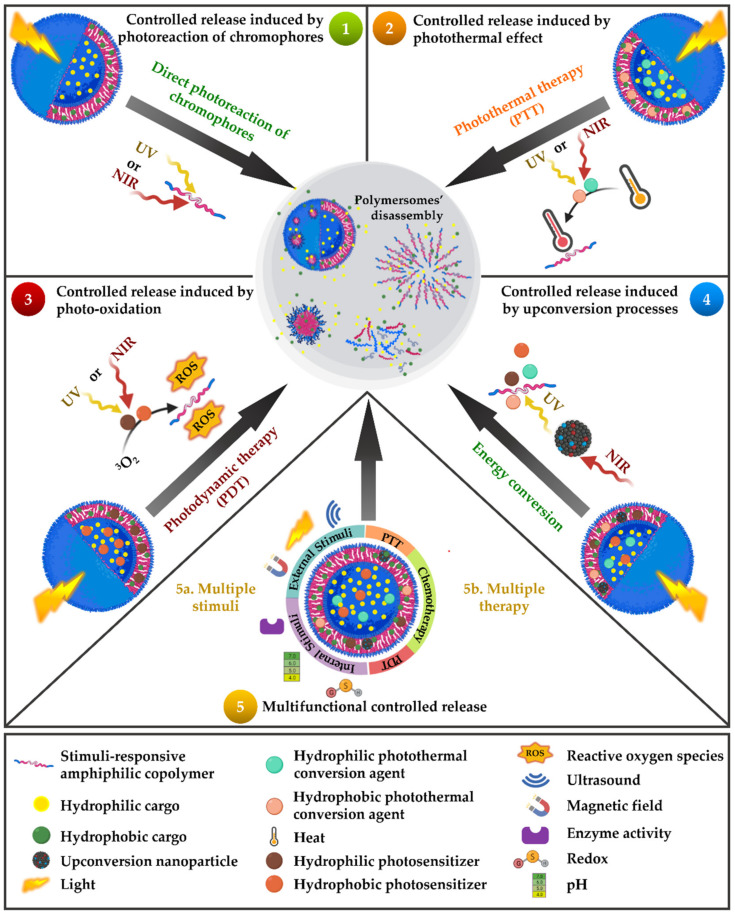
Schematic illustration of the therapeutic agent release mechanism of the light-responsive polymersomes induced by light-mediated (**1**) photoreactions of chromophores, (**2**) the photothermal effect, (**3**) photo-oxidation, (**4**) upconversion processes, and (**5**) multifunctional controlled-release.

**Figure 7 nanomaterials-12-00836-f007:**
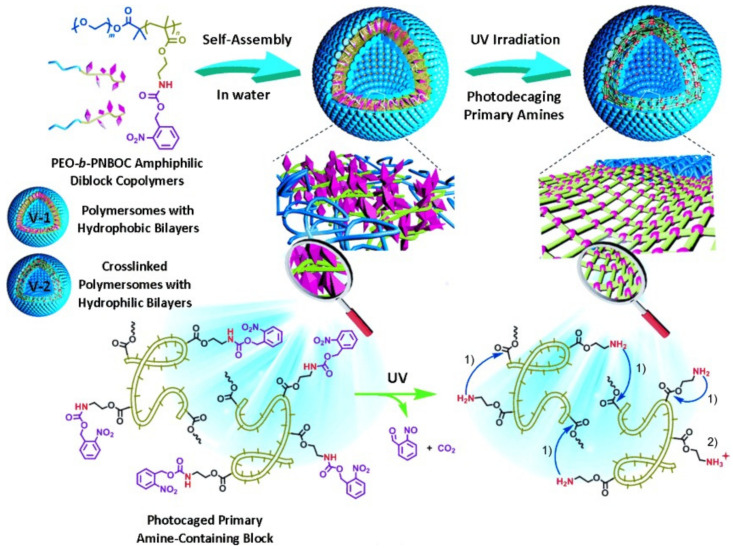
Design and mechanism of polymeric vesicles self-assembled from PEO_45_-*b*-PNBOC_30_ exhibiting concurrent phototriggered “traceless” crosslinking and vesicle membrane permeabilization. Reprinted with permission from [248]. Copyright © 2022, Wiley-VCH Verlag GmbH & Co. KGaA, Weinheim.

**Figure 8 nanomaterials-12-00836-f008:**
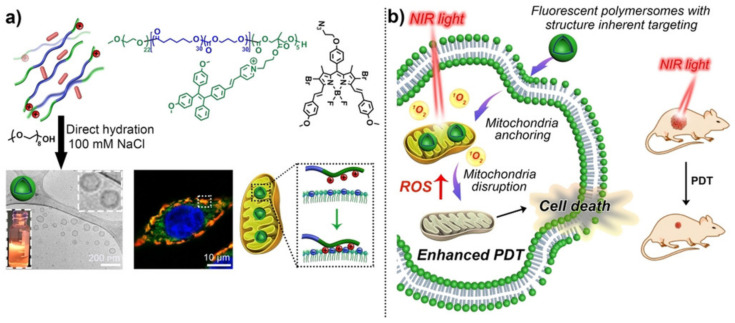
Assembly of biodegradable fluorescent polymersomes toward mitochondria-targeted PDT. (**a**) The polymer chemical structure and self-assembly into fluorescent polymersomes, and (**b**) the mechanism of action of fluorescent polymersomes that kill cancer cells in vitro and inhibit tumor growth in vivo. Reprinted with permission from [81]. Copyright © 2022, Angewandte Chemie International Edition published by Wiley-VCH GmbH.

**Figure 9 nanomaterials-12-00836-f009:**
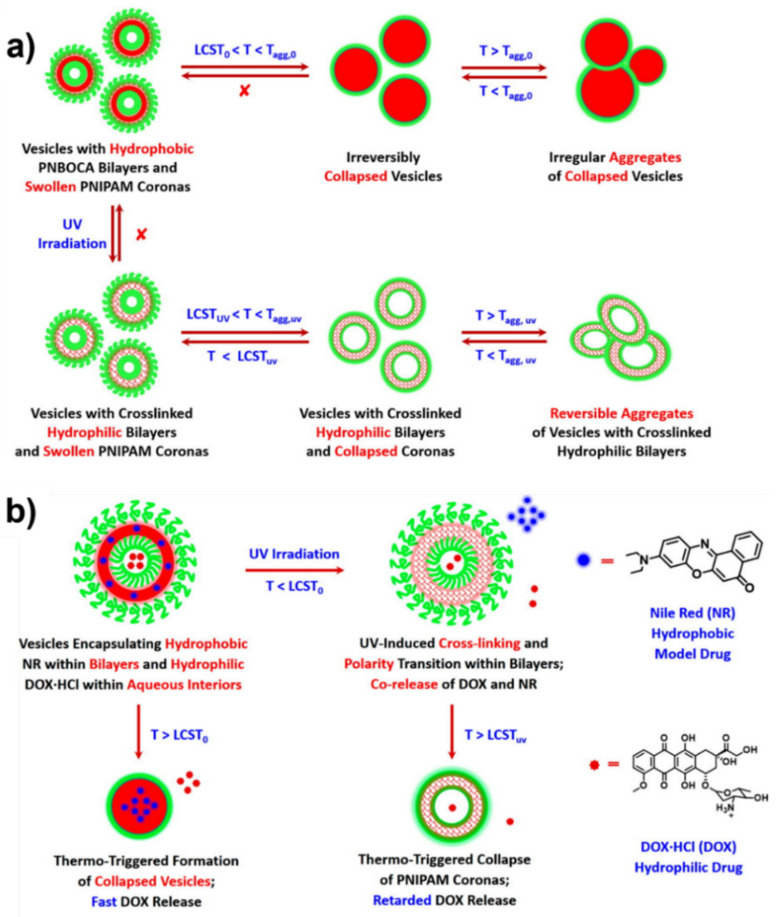
Schematic representation of (**a**) photo- and thermoresponsive polymersomes ensembled from PEO45-*b*-PCSSMA22, and (**b**) UV light- and temperature-regulated co-release of both hydrophilic DOX and hydrophobic NR payloads. Adapted with permission from [289]. Copyright © 2022, American Chemical Society.

**Figure 10 nanomaterials-12-00836-f010:**
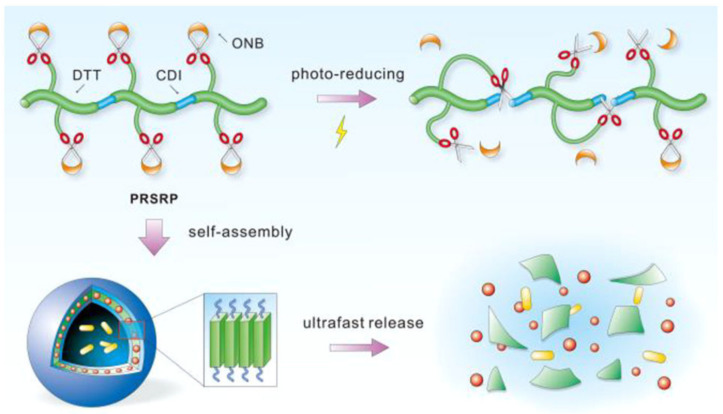
Schematic illustration of photo-responsive self-reducible polymers, self-assembly, and photo-reductive polymersome degradation. Reprinted with permission from [292]. Copyright © 2022, American Chemical Society.

**Figure 11 nanomaterials-12-00836-f011:**
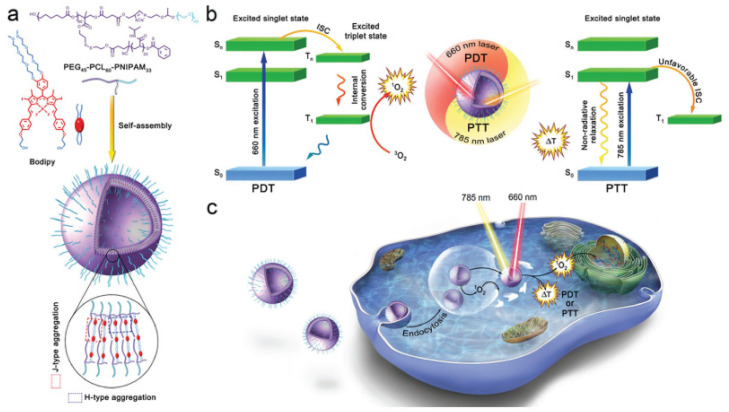
Schematic illustration of (**a**) photoconversion-tunable fluorophore polymersomes of both J-type and H-type BODIPY aggregates, (**b**) wavelength-dependent NIR-induced cancer therapy, including PDT under 660 nm irradiation or PTT under 785 nm irradiation, and (**c**) NIR-activation of polymersomes for PDT and PTT in the cell. Reprinted with permission from [295]. Copyright © 2022 WILEY-VCH Verlag GmbH & Co. KGaA, Weinheim.

**Figure 12 nanomaterials-12-00836-f012:**
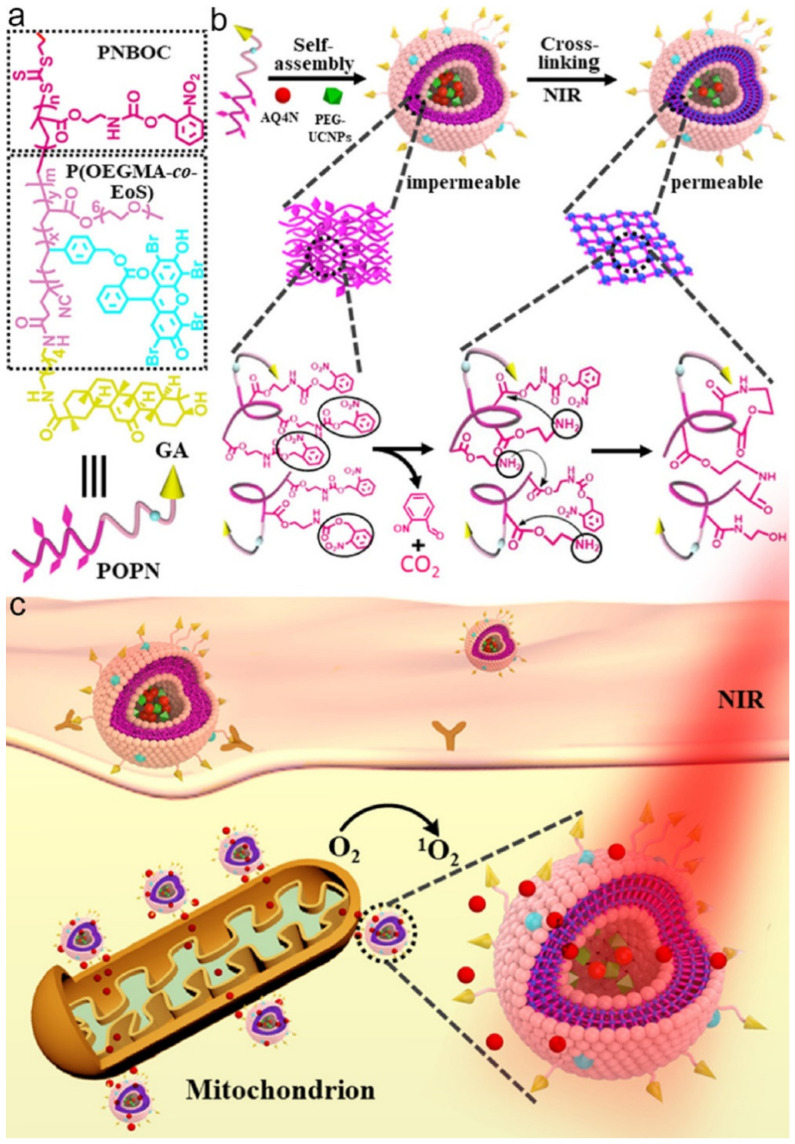
Schematic illustration of (**a**) amphiphilic copolymer POPN, (**b**) polymersome self-assembly with a permeability conversion mechanism, and (**c**) mitochondrion location and the intracellular delivery of AQ4N and PDT activated by NIR. Reprinted with permission from [303]. Copyright © 2022, American Chemical Society.

**Figure 13 nanomaterials-12-00836-f013:**
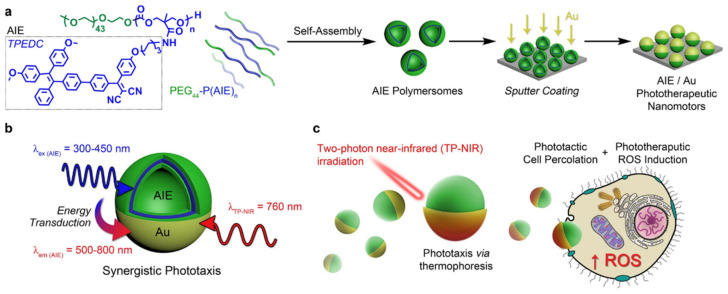
Schematic representation of AIE-transduced phototherapeutic nanomotors. (**a**) Biodegradable copolymers comprising AIEgenic TPEDC moieties, self-assembly into polymersomes, and sputter coating with gold to give hybrid AIE/Au nanomotors, (**b**) synergistic motion underpins nanomotor performance using two-photon NIR (TP-NIR, 760 nm) and is indirectly activated via energy transduction from the AIE polymersome (excited by TP-NIR at 380 nm), and (**c**) TP-NIR activation of nanomotors triggers dual behavior: enhancing cellular interactions and uptake alongside phototherapeutic ROS generation for highly localized cell toxicity. Reprinted with permission from [278]. Copyright © 2022, Shoupeng Cao et al.

**Figure 14 nanomaterials-12-00836-f014:**
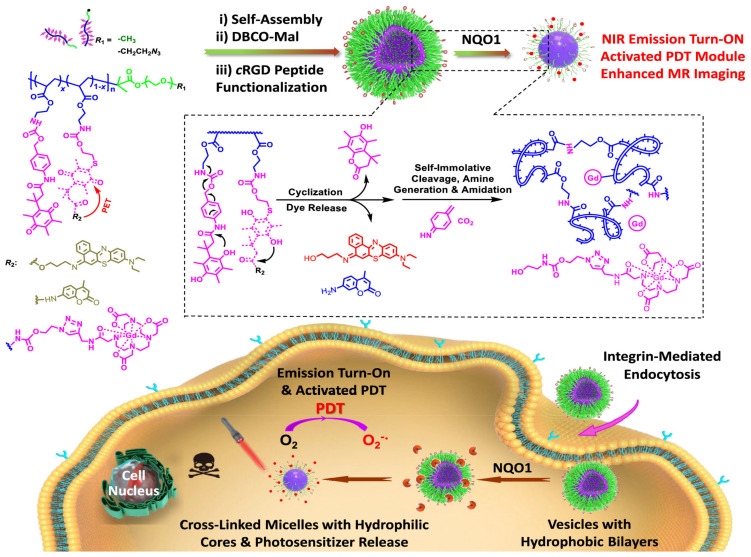
Tumor-cell-targeted NQO1 enzyme-responsive polymeric vesicles were self-assembled from amphiphilic copolymer PSs (coumarin and Nile blue) in the hydrophobic block. The mechanism of fluorescence emission, PDT, and MR imaging are activable by light. Reprinted with permission from [310]. Copyright © 2022, American Chemical Society.

**Table 1 nanomaterials-12-00836-t001:** Examples of amphiphilic copolymers reported for polymersome self-assembly.

Amphiphilic Copolymer Type	Polymersome Building Blocks	Ref.
Block copolymer	PS-*b*-PAA	[77]
	PEE_37_-*b*-PEO_40_	[78]
	PNBA-*b*-PDMA	[80]
	PEG-P(CLgTMC)	[81]
Graft copolymer	PHOU-*g*-PEG	[84]
	PEG/EAB–PPPs	[85]
	PGA-*g*-(PCL-*b*-PEO)	[86]
Random copolymer	NALGA-NADA	[90]
	PPAPE	[91]
Alternate copolymer	P(MF-alt-VBP)	[101]
Dendronized copolymer	PS-dendr-(NH_2_)_8_	[108]
	PEG-bis-MPA	[109,110]
Gradient copolymer	MPOx	[122]
	P(AA-grad-TFEMA)	[123]

Abbreviations: MPOx: Poly(2-methyl-2-oxazoline)-*g*-poly(2- phenyl-2-oxazoline); NALGA-NADA: *N*-acroloyl-l-glutamic acid and N-acroloyl-dodecyl amine; P(AA-grad-TFEMA): acrylic acid-gradient-2,2,2-Trifluoroethyl methacrylate; PEE_37_-b-PEO_40_: poly(ethylene oxide)-*b*-poly(ethyl ethylene); PEG-bis-MPA: Poly(ethylene glycol)-azobenzene-containing dendrons based on 2,2-bis(hydroxymethyl)propionic acid; PEG/EAB–PPPs: methoxypoly(ethylene glycol) and ethyl-p-aminobenzoate side groups on poly(dichlorophosphazene) backbone; PEG-P(CLgTMC): poly(ethylene glycol)-*b*-poly(caprolactone-gradient-trimethylene carbonate); PGA-*g*-(PCL-*b*-PEO): poly(glycerol adipate)-*g*-(poly(3-caprolactone)-*b*-poly(ethylene oxide)); PHOU-*g*-PEG: poly(3-hydroxyoctanoate-co-3-hydroxyundecanoate)-*g*-poly(ethylene glycol); P(MF-alt-VBP): poly(ethylene glycol) functionalized styrene and fatty acid attached maleimide monomers; PNBA-b-PDMA: poly(o-nitrobenzyl acrylate)-*b*-poly(*N*,*N*′-dimethyl acrylamide); PPAPE: poly(2-[4-(phenylazo)phenoxy]ethyl acrylate-co-acrylic acid); PS-b-PAA: polystyrene-*b*-poly(acrylic acid); PS-dendr-(NH_2_)_8_: polystyrene with poly(propylene imine) dendrimers.

**Table 2 nanomaterials-12-00836-t002:** Design of polymersomes for controlled release induced by the photoreaction of chromophores.

Polymersome Building Blocks	Light-Activable Moieties	Light Stimulus Wavelength (nm)	Assembly Method	Therapeutic Cargo	Size (nm)	Cell Line/Biological Model	Ref.
PEG-AuNPs-PNBA	NBA	365	Film rehydratation	DOX	210 ^a^	MDA-MB-435	[247]
PEO_45_-*b*-PNBOC_30_	ONB	365	Nanoprecipitacion	DOX	-	-	[248]
PEO-*b*-PSPA	SP	365/530	Nanoprecipitation	5-dFu	450 ^a^	-	[249]
PAMAM	Azide	365	Nanoprecipitation	OVA	121 ^a^	RAW264.7	[198]
PDMA-*b*-PNBA	ONB	365	Emulsion	DOX	-	-	[80]
C12NB	ONB	365	Nanoprecipitation	DOX	80–150 ^b^	HeLa	[250]
PRO and PEG-P(Asp-AP)	PRO	312	Vortex mixing	AsNEAT2	80 ^a^	A549	[251]

^a^ Determined by DLS, ^b^ Determined by TEM. Abbreviations: A549: human lung cancer cell line; AsNEAT2: nuclear-enriched abundant transcript 2 (NEAT2)-targeting ASO; AuNPs: gold nanoparticles; 5-dFu: 2′-deoxy-5-fluorouridine; DOX: doxorubicin; HeLa: human cervical carcinoma cell line; MDA-MB-435: breast cancer cell line; ONB: o-nitrobenzyl; OVA: ovalbumin; PAMAM: polypeptide-glycosylated poly-(amidoamine); PDMA-*b*-PNBA: poly(N,N’-dimethylacrylamide)-*b*-poly(o-nitrobenzyl acrylate); PEG: poly(ethylene glycol); PEG-P(Asp-AP): poly(ethylene glycol)-*b*-poly[N-(5-aminopentyl)-α,β-aspartamide]; PEO_45_-*b*-PNBOC_30_: poly(ethylene oxide)_45_-*b*-poly(2-nitrobenzyloxycarbonylaminoethyl methacrylate)_30_; PEO-*b*-PSPA: poly(ethylene oxide)-*b*-PSPA; PNBA: poly(o-nitrobenzyl acrylate); PRO: anionic photo-reactive oligodeoxynucleotides; RAW264.7: murine leukemic monocyte macrophage cell line; SP: spiropyran.

**Table 3 nanomaterials-12-00836-t003:** Design of polymersomes for controlled release induced by the photothermal effect.

Polymersome Building Blocks	Light-Activable Moieties	Light Stimulus Wavelength (nm)	Assembly Method	Therapeutic Cargo	Size (nm)	Ref.
PEG-AuNRs-PMMA	AuNRs	785	Film rehydratation	AuNRs	200 ^b^	[257]
PNIPAM-*b*-PAZO	AZO	365	Nanoprecipitation	MN	5 µm ^b^	[258]
PBD_35_-*b*-PEO_20_	AuNPs	532 nm	Gel-assisted rehydration	AuNPs	71 ^a^	[189]

^a^ Determined by DLS, ^b^ Determined by TEM. Abbreviations: AuNPs: gold nanoparticles; AuNRs: gold nanorods; AZO: azobenzene; MN: Fe_3_O_4_ magnetic nanoparticles; PBD_35_-b-PEO_20_: poly(butadiene)_35_-*b*-poly(ethylene oxide)_20_; PEG-AuNPs-PMMA: poly(ethylene glycol)-gold nanorods-poly(methyl methacrylate); PNIPAM-*b*-PAZO: poly(*N*-isopropyl acrylamide)-*b*-poly{6-[4-(4-methoxyphenylazo)phenoxy] hexylacrylate}.

**Table 4 nanomaterials-12-00836-t004:** Design of polymersomes for controlled release induced by photo-oxidation.

Polymersome Building Blocks	Light-Activable Moieties	Light Stimulus Wavelength (nm)	Assembly Method	Therapeutic Cargo	Size (nm)	Cell Line/Biological Model	Ref.
PPLA	Porphyrin	420	Oil-in-water solvent evaporation	Porphyrin	52	HeLa	[265]
PMOXA–PDMS–PMOXA	Rose Bengal	543	Film rehydration	Rose Bengal	R_g_: 105 ^a^ R_h_: 110 ^a^	HeLa	[196]
PEG_45_-(PAsp)_75_ homo-P(Asp-AP)_82_	AlPcS_2a_	400–700	Vortex mixing	AlPcS_2a_	106 ^a^	A549	[266]
CC-PAMAM Thioketal moiety-PEG	Ce6	660	Nanoprecipitation	Ce6	187.5 ^a^ 155.2 ^b^	BxPC-3 BALB/c nude mice	[267]
PEG-PCL-PEI	PheoA	670	Nanoprecipitation	PheoA	200–280 ^a^ 200 ^b^	HeLa	[268]
PEG-P(CLgTMC)	BODIPY	606	Direct hydratation	BODIPY	-	A549, HeLa, HepG2, and 3T3 A549 tumor-bearing nude mice	[81]

^a^ Determined by DLS, ^b^ Determined by TEM. Abbreviations: A549: adenocarcinomic human alveolar basal epithelial cell line; AlPcS_2a_: Al(III) phthalocyanine chloride disulfonic acid; BODIPY: boron-dipyrromethene; BxPC-3: human pancreatic cancer cells; CC-PAMAM: poly(amidoamine) dendrimer conjugating chlorin e6/cypate; Ce6: Chlorin e6; HeLa: human cervical carcinoma cell line; HepG2: human hepatocyte carcinoma cell line; Homo-P(Asp-AP)_82_: homocatiomer poly([5-aminopentyl]-α,β- aspartamide)_82_; pDNA: endotoxin-free GFP-encoding plasmid DNA; PEG: poly(ethylene glycol); PEG_45_-(Pasp)_75_: poly(ethylene glycol)_45_-poly(α,β-asparticacid)_75_; PEG-PCL-PEI: poly(ethylene glycol)-polycaprolactone-poly(ethylene imine); PEG-P(CLgTMC): poly(ethylene glycol)-b-poly(caprolactone-gradient-trimethylene carbonate); PheoA: pheophorbide-a; PMOXA–PDMS–PMOXA: poly(2-methyloxazoline)-poly(dimethylsiloxane)-poly-(2-methyloxazoline); PPLA: porphyrin–polylactide conjugates; R_g_: radius of gyration; R_h_: hydrodynamic radius; 3T3: healthy fibroblast cell line.

**Table 5 nanomaterials-12-00836-t005:** Design of polymersomes for release controlled by multiple stimuli.

Stimuli	Polymersomes Building Blocks	Light-Activable Moieties	Light Stimulus Wavelength (nm)	Assembly Method	Therapeutic Cargo	Size (nm)	Cell Line/Biological Model	Ref.
Light Redox	PNBC-*b*-PEO	ONB Thiol	365	Film rehydratation	DOX	83 ^a^	HeLa	[287]
Light pH	β-CD-AZO-ACE⊂ α-CD	AZO Acetal moiety	365	Nanoprecipitation	DOX	40 ^a^ 88.5 ^b^	-	[288]
Light Thermo	PNIPAM_31_-*b*-PNBOCA_53_	ONB PNIPAM	365 808	Nanoprecipitation	DOX	-	-	[289]
Light Redox	PEO_45_-*b*-PCSSMA_22_	Coumarin Disulfide linkage	430	Nanoprecipitation	DOX TR-dextran	-	-	[290]
Light pH	PCL-Azo + WP6	AZO WP6	365 435	Nanoprecipitation	DOX		HepG2	[291]
Light Redox	ONB-DTT-CDI-MPEG	ONB Disulfide linkage	365	Solvent exchange	DOX	R_g_:35.5 ^a^ R_h_:34.8 ^a^ 90 ^a^	MCF-7	[292]
Light Redox pH	PEO_45_-*b*-P(NCMA_0.55_- *co*-DPA_0.45_)_29_; PEO_45_-*b*-PNCMA_17_-*b*-PDPA_21_; PEO_45_-*b*-P(DCMA_0.45_-*co*-PDPA_0.55_)_33_	ONB Disulfide likage Carbamate Tertiary amine	365	Nanoprecipitation	Gem DOX 5-FU Calcein	540–770 ^a^	-	[207]
Light Redox	PCL-ONB-SS-PMAA	UCNP ONB Disulfide linkage	980	Double emulsion	DOX	-	A549, CR-5802, HEL-299 A549 tumor bearing mice	[293]

^a^ Determined by DLS, ^b^ Determined by TEM. Abbreviations: A549: adenocarcinomic human alveolar basal epithelial cell line; AZO: azobenzene; β-CD-AZO-ACE⊂α-CD: inclusion complex of β-cyclodextrin, azobenzene, acetal and α cyclodextrin; CR-5802: lung cancer cell line; DOX: doxorubicin; DCMA: disulfide-caged carboxyl monomer; DPA: carbamate linkage; 5-FU: 5-Fluouracil; Gem: gemcitabine; HEL-299: human fetal lung fibroblast cell line; HeLa: human cervical carcinoma cell line; HepG2: human hepatocyte carcinoma cell line; MCF-7: breast cancer cell line; NCMA: 2-nitrobenzyl ester-photocaged carboxyl monomer; ONB: o-nitrobenzyl; ONB-DTT-CDI-MPEG: O-nitrobenzyl protected dithiothreitol-l-cystine dimethyl ester diisocyanate-methoxyl poly(ethylene glycol); PCL-Azo+WP6: AZO ended functionalized poly(e-caprolactone) + water soluble pillar[6]arene; PCL-ONB-SS-PMAA: poly(ε-caprolactone)-o-nitrobenzyl-SS-poly(methacrylic acid); PEO: poly(ethylene oxide); PEO_45_-*b*-PCSSMA_22_: poly(ethylene oxide)_45_-*b*-poly(coumarin-based disulfide-containing monomer)_22_; PNBC-*b*-PEO: polypeptide poly(S-(o-nitrobenzyl)-l-cysteine)-*b*-poly(ethylene oxide); PNIPAM: poly(*N*-isopropylacrylamide); PNIPAM_31_-b-PNBOCA_53_: poly(N-isopropylacrylamide)-*b*-poly(2-((((2-nitrobenzyl)-oxy)carbonyl)amino)ethyl acrylate); R_g_: radius of gyration; R_h_: hydrodynamic radius; TR-dextran: texas red-labeled dextran; UCNP: upconversion nanoparticles; WP6: water soluble pillar[6]arene.

**Table 6 nanomaterials-12-00836-t006:** Design of polymersomes for the controlled release of multiple therapies.

Therapies	Polymersome Building Blocks	Light-Activable Moieties	Light Stimuli Wavelength (nm)	Assembly Method	Therapeutic Cargo	Size (nm)	Cell Line/Biological Model	Ref.
Chemotherapy PTT	mPEG-PCL	AuNRs	808	Double emulsion	DOX	208 ^a^ 175 ^b^	C26 Mice bearing C26 tumors	[294]
PTT PDT	PEG_45_-PCL_60_-PNIPAM_33_	BODIPY	660 785	Ultrasonication	-	127.3 ^a^ 72.5 ^b^	4T1 Mice bearing 4T1 tumors	[295]
Chemotherapy PTT	PCL_8000_-PEG_8000_-PCL_8000_	ICG	808	Film rehydration	DOX	208.1 ^a^	4T1-Luc	[296]
Chemotherapy PTT	PEO-*b*-PMALA PNBOC-*b*-PMALA	ONB	410	Nanoprecipitation	DOX PTX AuNPs	194 ^a,b^	HepG2	[297]
Chemotherapy PTT	PPS-PEG	CR780 dye	808	Film rehydration	DOX	100 ^a^	U87MG Mice bearing U87MG tumors	[298]
Chemotherapy PTT	PEG-b-PLA/DOPC	AuNRs	808	Double emulsion	DOX Docetaxel Rapamycin Afatinib	286 ^b^	SKBR-3/AR HER2-positive breast cancer mouse model	[299]
Chemotherapy PTT PDT	mPEG-*b*-PBAC-*b*-mPEG	IR-780 PBAC	808	Nanoprecipitacion	DOX	204 ^b^	4T1, CT26 4T1 tumor bearing mice	[300]
Chemotherapy PDT	PPS_20_-*b*-PEG_12_	ZnPc	660	Solvent dispersion	DOX	150 ^a^	A375 Nude mice bearing A375	[301]
PTT PDT	PEG-PCL	ICG AuNRs	785	Ultrasonication	-	238 ^b^	PC3 Nude mice bearing PC3	[302]
Chemotherapy PDT	b-(AZO-grafted Dex)-b((MMA)-r-(β-CDAc)-r-(PorAc))	AZO Porphyrin	365	Nanoprecipitation	Quercetin 5-FU	200 ^b^	-	[200]
Chemotherapy PDT	P(OEGMA-co-EoS)-*b*-PNBOC	UCNPs ONB	980	Nanoprecipitation	AQ4N	234 ^a^ 209 ^b^	HepG2	[303]

^a^ Determined by DLS, ^b^ Determined by TEM. Abbreviations: A375: human melanoma cell line; AQ4N: banoxantrone dihydrochloride; AuNPs: gold nanoparticles; AuNRs: gold nanorods; AZO: azobenzene; b-(Azo-grafted Dex)-b((MMA)-r-(β-CDAc)-r-(PorAc)):b-(azobenzene-grafted Dextran)-b(methyl methacrylate)-r-(mono-methacrylate modified beta-cyclodextrin)-r-(porphyrin acrylate)); BODIPY: boron-dipyrromethene; C26: mouse colon carcinoma cell line; CR780: dye croconaine; CT26: murine colorectal carcinoma cell line; DOX: doxorubicin; 5-FU: 5-Fluouracil; HeLa: human cervical carcinoma cell line; HepG2: human hepatocyte carcinoma cell line; ICG: indocynanine green; mPEG-*b*-PBAC-*b*-mPEG: monomethyl poly(ethylene glycol)-*b*-poly(β-aminoacrylate)-*b*- monomethyl poly(ethylene glycol); mPEG-PCL: monomethyl poly(ethylene glycol)-poly(ε-caprolactone); ONB: o-nitrobenzyl; PBAC: poly(β-aminoacrylate); PC3: human prostate cancer bone metastases; PCL_8000_-PEG_8000_-PCL_8000_: poly(ε-caprolactone)_8000_-poly(ethylene glycol)_8000_-poly(ε-caprolactone)_8000_; PDT: photodynamic therapy; PEG-PCL: poly(ethylene glycol)-poly(ε-caprolactone); PEG_45_-PCL_60_-PNIPAM_33_: poly(ethylene glycol)_45_-poly(ε-caprolactone)_60_-poly(*N*-Isopropylcrylamide)_33_; PEO-*b*-PMALA: poly(ethylene oxide)-*b*-poly(2-(methacryloyloxy) ethyl 5-(1,2-dithiolan-3-yl)pentanoate); PNBOC-*b*-PMALA: poly(2-((((2-nitrobenzyl)oxy)carbonyl)amino)ethyl methacry-late)-*b*-poly-(2-(methacryloyloxy)ethyl 5-(1,2-dithiolan-3-yl)pentanoate)); PPS_20_-*b*-PEG_12_: poly(propylene sulfide)_20_-*b*-poly(ethylene glycol)_12_; PPS-PEG: poly(propylene sulfide)-poly(ethyle glycol); P(OEGMA-co-EoS)-b-PNBOC: poly[oligo(ethylene glycol) monomethyl ether methacrylate-co-eosin Y]-*b*-poly(2-nitrobenzyl oxycarbonyl aminoethyl methacrylate); PTT: photothermal therapy; PTX: paclitaxel; 4T1: murine mammary carcinoma cell line; U87MG: human primary glioblastoma cell line; UCNP: upconversion nanoparticles; ZnPc: zinc phthalocyanine.

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
