# Peer review of "Light-Triggered Polymersome-Based Anticancer Therapeutics Delivery"

_nanomaterials, 2022, doi:10.3390/nano12050836_

Round 1

Reviewer 1 Report

Elisa Hernández Becerra and colleagues summarize the state-of-art light-triggered polymersome-based anticancer therapeutics delivery. The references in this review article were well-selected and well-presented. Additionally, the structure and flow of this article are very organized and clear. I have very few comments or concerns. I recommend this paper be accepted.

  1. I found this work did a superb job refining almost all the aspects of the light-triggered polymersome. Specifically, the authors highlighted the synthesis, functionalization, and characterization of polymersomes. Although these experimental results are helpful, I recommend the authors also highlight the existing in silico modeling methods for polymersomes. The theoretical modeling typically assists and guides the experimental design. Therefore, it could be instructive to summarize the up-to-date reference on the simulation methods.
  2. Page 2. Section “2. Influence of biological barriers on Drug Delivery Systems”. The authors highlighted multiple nanoparticle drug delivery systems. In the most recent decades, a new nanocarrier, Metal-Organic Framework (MOFs) has emerged as an ideal drug delivery system. To create an even broader audience for this review, I recommend the authors also include references about the MOFs and show the application of MOFs in this section.

Author Response

Response to Reviewer 1

Title: Light-triggered polymersome-based anticancer therapeutics delivery

Manuscript ID: nanomaterials-1596855

Type of manuscript: Review

Authors: Elisa Hernández Becerra, Jennifer Quinchia, Cristina Castro, Jahir Orozco*

Here we provide a point-by-point response to all questions and comments raised by Reviewer 1. Moreover, the new text and corrections are highlighted in the revised manuscript.

General comment:

Elisa Hernández Becerra and colleagues summarize the state-of-art light-triggered polymersome-based anticancer therapeutics delivery. The references in this review article were well-selected and well-presented. Additionally, the structure and flow of this article are very organized and clear. I have very few comments or concerns. I recommend this paper be accepted.

We gratefully acknowledge reviewer 1 for his/her favorable opinion on our work and the suggestions and recommendations for improving the manuscript.

Specific comments:

1. I found this work did a superb job refining almost all the aspects of the light-triggered polymersome. Specifically, the authors highlighted the synthesis, functionalization, and characterization of polymersomes. Although these experimental results are helpful, I recommend the authors also highlight the existing in silico modeling methods for polymersomes. The theoretical modeling typically assists and guides the experimental design. Therefore, it could be instructive to summarize the up-to-date reference on the simulation methods.

Based on this suggestion, some references related to computational and modeling polymersomes were included in section 3.2 (Prog. Polym. Sci. 2007, 32, 838–857; Macromolecules 2021, 54, 9258–9267; Soft Matter 2020, 16, 3234–3244; Sensors 2019, Vol. 19, Page 5266 2019, 19, 5266; J. Phys. Chem. B 2005, 109, 17708–17714; Phys. Chem. Chem. Phys. 2020, 22, 24934–24942; Macromol. Rapid Commun. 2009, 30, 267–277; J. Lab. Autom. 2013, 18, 34–45; Langmuir 2017, 33, 10084–10093; Nanomedicine Nanotechnology, Biol. Med. 2014; Faraday Discuss. 2005, 128, 355–361; Nanomater. 2018, Vol. 8, Page 763 2018, 8, 763).

2. Page 2. Section “2. Influence of biological barriers on Drug Delivery Systems”. The authors highlighted multiple nanoparticle drug delivery systems. In the most recent decades, a new nanocarrier, Metal-Organic Framework (MOFs) has emerged as an ideal drug delivery system. To create an even broader audience for this review, I recommend the authors also include references about the MOFs and show the application of MOFs in this section.

According to reviewer 1 suggestion, we include information about metal-organic framework (MOF) nanocarriers for drug delivery in section 2.

Reviewer 2 Report

Comments to the Authors:

This is an interesting and overall well-written paper, this manuscript describes current light-triggered polymersomes evaluated for cancer therapeutics agents' delivery and anticancer therapy. Firstly, it introduces the influence of biological barriers on drug delivery systems. And then it discusses species and synthesis of some polymersomes for anticancer therapeutics delivery. Lastly, it introduces the mechanism of light-responsive polymersomes for anticancer therapeutics delivery. It will be a solid contribution to the Nanomaterials and will certainly appeal to many of its readers. I address some of the main issues with the manuscript in the next few paragraphs. It is recommended that this manuscript be published in Nanomaterials after completing minor revision.

  1. Figure 1 cannot directly reflect the Amphiphilic copolymer types and synthesis, please changing the picture or adding pictures and tables.
  2. Adjusting the size of the pictures, focusing on the overall layout. Figure 2, trying to make pictures adapt the paper size.
  3. Please enriching the first half of the content, adding relevant pictures. Relevant pictures can make the content of the article concise and easy to understand.
  4. The introduction only introduces the application of polymersomes for cancer therapeutics agents' delivery and anticancer therapy. It is too brief, some other methods, such as supramolecular design, for cancer therapeutics agents' delivery and anticancer therapy should be introduced. So the following recently published important related papers should be cited: Chem. Soc. Rev. 2017, 46, 7021; Chem. Soc. Rev. 2021, 50, 2839.

Author Response

Response to Reviewer 2

Title: Light-triggered polymersome-based anticancer therapeutics delivery

Manuscript ID: nanomaterials-1596855

Type of manuscript: Review

Authors: Elisa Hernández Becerra, Jennifer Quinchia, Cristina Castro, Jahir Orozco*

Here we provide a point-by-point response to all questions and comments raised by Reviewer 2. Moreover, the new text and corrections are highlighted in the revised manuscript.

General comment:

This is an interesting and overall well-written paper, this manuscript describes current light-triggered polymersomes evaluated for cancer therapeutics agents' delivery and anticancer therapy. Firstly, it introduces the influence of biological barriers on drug delivery systems. And then it discusses species and synthesis of some polymersomes for anticancer therapeutics delivery. Lastly, it introduces the mechanism of light-responsive polymersomes for anticancer therapeutics delivery. It will be a solid contribution to the Nanomaterials and will certainly appeal to many of its readers. I address some of the main issues with the manuscript in the next few paragraphs. It is recommended that this manuscript be published in Nanomaterials after completing minor revision.

We gratefully acknowledge reviewer 2 for his/her favorable opinion on our work and the suggestions and recommendations for improving the manuscript.

Specific comments:

1. Figure 1 cannot directly reflect the Amphiphilic copolymer types and synthesis, please changing the picture or adding pictures and tables.

We thank the reviewer for his/her comment and would like to clarify that Figure 3 (before it was called Figure 1) schematizes different ways of forming a polymersome membrane from amphiphilic copolymers (block, graft, random, alternate, and dendronized) but not their synthesis. This Figure was designed based on previous reports, Biomacromolecules 2020, 21, 1327−1350; Current Pharmaceutical Design 2011, 17(1), 65-79; Journal of Materials Chemistry 2009, 19(22), 3576; Chemical Society Review, 2018, 47, 8572-8610; Polymer Chemistry, 2014, 5, 4069-4075. In this context, although the Figure doesn't reflect the amphiphilic copolymer synthesis, it does amphiphilic copolymer the types. Comments related to the synthesis are in sections 3.1.1.-3.1.6 , pg 8-12.

2. Adjusting the size of the pictures, focusing on the overall layout. Figure 2, trying to make pictures adapt the paper size.

Figure 5 (Light-activable moieties according to the release mechanism of the light-responsive polymersomes) and Figure 6 (Schematic illustration of the therapeutic agent release mechanism of the light-responsive polymersomes induced by light-mediated photoreactions of chromophores, photothermal and photo-oxidation) were adjusted to the paper size. It was impossible to adapt Figure 4 (before it was called Figure 2) to horizontal paper size by the resolution of the Figure.

3. Please enriching the first half of the content, adding relevant pictures. Relevant pictures can make the content of the article concise and easy to understand.

We added two relevant Figures in section 2, i.e., Figure 1 (Biochemical and physicochemical properties of NPs for enhanced DDSs) and Figure 2 (Pathways for cellular uptake mediated by endocytic). Moreover, we added Table 1 in section 3.

4. The introduction only introduces the application of polymersomes for cancer therapeutics agents' delivery and anticancer therapy. It is too brief, some other methods, such as supramolecular design, for cancer therapeutics agents' delivery and anticancer therapy should be introduced. So the following recently published important related papers should be cited: Chem. Soc. Rev. 2017, 46, 7021; Chem. Soc. Rev. 2021, 50, 2839.

 The introduction was extended, and two proposed references (Chem. Soc. Rev. 2017, 46, 7021; Chem. Soc. Rev. 2021, 50, 2839) were included.
